# Effect of Low Temperature on Photosynthetic Physiological Activity of Different Photoperiod Types of Strawberry Seedlings and Stress Diagnosis

**Nan Jiang** **, Zaiqiang Yang \*, Hanqi Zhang, Jiaqing Xu and Chunying Li**

School of Applied Meteorology, Nanjing University of Information Science & Technology, Nanjing 210044, China; 202211080002@nuist.edu.cn (N.J.)
* Correspondence: yzq@nuist.edu.cn; Tel.: +86-134-0196-3706

**Abstract:** During the early growth stage of plants, low temperatures can alter cell permeability, reduce photosynthetic capacity, and have adverse effects on crop growth, development, and yield. Different strawberry cultivars have varying cold tolerance. In this study, we investigated the changes in cell permeability and photosynthetic activity of short-day and long-day types of strawberry cultivars under varying degrees of low-temperature stress, and evaluated the extent of cellular damage using photosynthetic and chlorophyll fluorescence parameters. The experiment utilized short-day strawberry cultivars 'Toyonoka' and 'Red Face', and long-day strawberry cultivars 'Selva' and 'Sweet Charlie' seedlings. Low-temperature treatments were set at $-20$, $-15$, $-10$, $-5$, 0, 5, and 10 °C for 12 h. The research demonstrated that short-day strawberries had greater tolerance to low temperatures, and all four strawberry cultivars began to experience low-temperature stress when the temperature was below 5 °C. A temperature range of 0 to $-10$ °C played a crucial role in causing severe cold damage to the strawberries. The low-temperature stress levels were constructed based on electrolyte leakage, with photosynthetic physiological characteristics serving as references. The study proves that the photosynthetic and chlorophyll fluorescence parameters can serve as effective probes for diagnosing low-temperature stress in strawberry seedlings, and their combination provides higher accuracy in identifying stress levels than any single type of parameter.

**Keywords:** Fragaria × ananassa Duch; long-day varieties; low-temperature stress; non-destructive diagnosis; OJIP transient; photosynthesis; relative electrical conductivity; seedling stage; short-day varieties; SIF



## 1. Introduction

Strawberry (*Fragaria × ananassa* Duch.), a perennial herbaceous plant of the Rosaceae family, is one of the most important crops widely cultivated worldwide. China's strawberry planting and production scale accounts for more than one-third of the global total [1]. It is rich in antioxidants, and has a beautiful fragrance and flavor, making it widely popular among consumers [2]. Based on the differences in photoperiod and fruiting period, strawberries can be divided into short-day and long-day types. Short-day strawberries, also known as seasonal strawberries, are sensitive to photoperiod, and grow and produce the best yields in environments with no more than 12 h of sunlight. If the light duration is too long, flower bud formation will be affected. Long-day strawberries, also called everbearing strawberries, require at least 12 h of sunlight for growth and development. At suitable temperatures, everbearing strawberries can grow and bear fruit all year round without a clear dormancy period [3].

In recent years, global climate change has been intensifying, leading to a significant increase in extreme weather events, including extremely low temperatures, which are expected to become even more frequent in the future [4]. Strawberries are a warm-loving crop that is highly sensitive to low temperatures. The optimal temperature range for

their growth is between 15 and 25 °C [5,6]. Strawberry seedlings have underdeveloped growing points and new leaves, making them more vulnerable to cold snaps and low temperatures. Exposure of strawberries to excessively low temperatures on seedlings induces depolymerization of microtubules and microfilaments [7,8], and has a sustained impact on growth and production [9,10].

Photosynthesis is the most fundamental life activity of plants, serving as the fundamental source of organic matter and energy, and photosynthetic organs are the cold-sensitive parts of plants [10,11]. Photosynthetic parameters can be used to describe the efficiency of a plant leaf in utilizing photosynthetically active radiation (PAR) and the rate of photosynthesis [12]. The impact of low temperature on the photosynthetic performance of thermophilic crops outweighs that on winter plants [13]. Transient chlorophyll fluorescence induction kinetics curve (OJIP transient) and its parameters [14–16], as well as solar-induced chlorophyll fluorescence (SIF) [17,18], can reveal the effects of biotic or abiotic stress on plant photosynthesis. This makes chlorophyll fluorescence, whether induced artificially or by sunlight, an important factor in assessing photosynthesis [19] or cold tolerance diagnosis of crops [20,21]. Hence, these indicators can serve as non-destructive probes to reflect the photosynthetic capacity and physiological activity of plants [22,23].

Measuring the electrolyte leakage rate of plant tissues can quantitatively assess the extent of damage to plants under stress [23]. Relative electrical conductivity (REC) reflects the degree of electrolyte leakage. Using the Logistic equation can fit the continuous changes in REC under different stress levels, providing information about the differences in stress tolerance among different cultivars. In practical agricultural production, however, the widespread use of this method for assessing and monitoring low-temperature stress would undoubtedly require a significant amount of time and manpower for sampling and experimentation, which could cause damage to crops and consequently affect economic benefits. Therefore, finding a method that can assess plant cell damage without the need for destructive sampling is of great significance.

To our knowledge, there are few studies that have investigated the changes in the photosynthetic physiological activity of strawberry seedlings under low-temperature stress, and no research has revealed the differences in cold resistance between short-day-type and long-day-type strawberry seedlings. Different photoperiodic strawberry varieties are suitable for growth in different seasons, which leads us to speculate that there may be differences in cold adaptation between short-day and long-day types. Thus, the study aims to investigate the variations and discrepancies in the photosynthetic physiological activity of short-day-type and long-day-type strawberry seedlings under low-temperature stress, and to establish a non-destructive diagnostic method for low-temperature stress degree during the seedling stage by key indicators derived from photosynthetic parameters and chlorophyll fluorescence characteristics.

## 2. Materials and Methods

### 2.1. Plant Materials

Different photoperiodic types of strawberry varieties widely cultivated in China were selected for the experiment, including short-day-type cultivars 'Toyonoka', 'Red Face', and long-day-type cultivars 'Selva', 'Sweet Charlie'. Seedlings with 4–6 fully formed and evenly colored true leaves, sturdy petioles, and a plant height of approximately 10–13 cm were selected and transplanted into plastic pots with dimensions of 15 cm (height) × 17.5 cm (upper diameter) × 14 cm (lower diameter), one plant per pot. The pots were filled with 3 L of substrate composed of vermiculite, perlite, peat, and garden soil in a ratio of 1:1:1:3 (v:v:v:v). Low-temperature experiments were conducted 20 days after transplantation. During the cultivation period, all test seedlings were subjected to uniform water and nutrient management until the start of the low-temperature treatment. The soil was fertilized with nitrogen (urea, 46% N, 150 kg/ha), phosphorus (calcium superphosphate, 12% $P_2O_5$, 200 kg/ha), and potassium (potassium sulfate, 52% $K_2O$, 250 kg/ha) as the basal fertilizer, without further fertilization since then. Water was supplied according to the

'5-point sampling method', with additional watering to saturation when the average soil moisture content reached approximately 60%. The watering was carried out between 16:00 and 18:00. Each treatment involved three strawberry plants, resulting in a total of 24 plants for each variety. A total of 96 strawberry seedlings were utilized for the four varieties with 96 separate pots.

### 2.2. Experimental Management and Treatment

The experiment was conducted at the Agrometeorological Experimental Station of Nanjing University of Information Science and Technology (NUIST) from October to November 2022. The strawberry seedlings were cultivated in an artificial climate chamber (PGC-FLEX, Conviron, Canada) with consistent light, temperature, and humidity. The photoperiod in the chamber was set to 12 h (7:00–19:00), and the photosynthetically active radiation (PAR) was set at 800 μmol m$^{-2}$ s$^{-1}$. The temperature and humidity in the chamber were set to simulate the hourly changes in the greenhouse microclimate in Nanjing (Figure 1) [24].

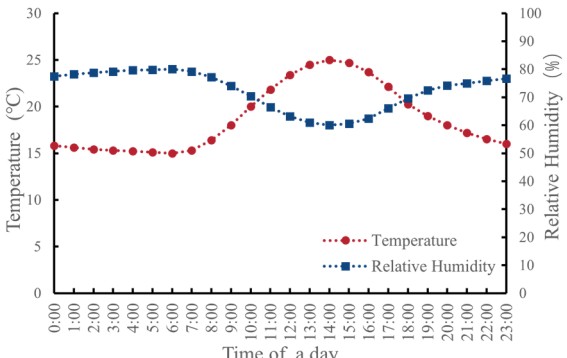

**Figure 1.** Hourly variation of temperature and relative humidity in artificial climate chamber. Constant temperature during each hour.

Seven low-temperature treatments were set to a continuous −20, −15, −10, −5, 0, 5, and 10 °C, with 20 °C as the control (CK). The treatments followed the photoperiod, with a duration of 12.5/11.5 h dark/light. The plants were moved into low-temperature incubator at 18:30 and taken out at the next 7:00. Due to limitations in observed variables and sample size, only two strawberry varieties could be treated at a time for low-temperature treatments, alternating between 'Toyonoka' and 'Selva' or 'Red Face' and 'Sweet Charlie'. Six plants (pots) were treated in each batch, and a total of 16 batches of stress treatments were conducted under controlled conditions. After stress treatment, plants were given a 2 h recovery period in the climate chamber to allow for acclimation to the environment. Observations began at 9:00 am, and the parameters of photosynthesis, transient chlorophyll fluorescence induction kinetics curve (OJIP transient), solar-induced chlorophyll fluorescence (SIF), and relative electrical conductivity (REC) were sequentially measured. Each observation was repeated using three plants per treatment.

### 2.3. The Methods of Measurement
#### 2.3.1. Semi-Lethal Temperature

Semi-lethal temperature (LT$_{50}$) was calculated using the Logistic equation based on REC of strawberry leaves. Two to three healthy functional leaves were selected per plant, wiped clean with deionized water, the main and secondary veins removed, and cut into pieces (d ≤ 0 mm). Weigh 1.5 g of chopped leaves and place them into a 25 mL conical flask. Add 20 mL of deionized water to the flask and vacuum the mixture for 20 min until the leaves settle at the bottom. The conductivity value C$_1$ was measured using a conductivity meter (DDS-307A, INESA, CHN). Then, the conical flask was placed in a

constant temperature water bath with boiling water for 15 min, and the conductivity value $C_2$ was recorded after it cooled down. REC was calculated using the following Equation (1).

$$REC = C_1/C_2 \times 100\% \tag{1}$$

In the equation, $C_1$ represents the REC at environment temperature of 20 °C; $C_2$ represents the REC after high-temperature water bath.

The general form of the Logistic equation is shown in Equation (2), where K is the saturation parameter of y, which, in this study, is REC and has a value of 100. The parameters a and b at the inflection point of the equation were calculated according to the method of Otieno et al. [25]. Then, $LT_{50}$ can be calculated based on Equation (3).

$$y = K/\left(1 + ae^{bx}\right) \tag{2}$$

$$LT_{50} = -\ln a/b \tag{3}$$

In the equation, y represents the value of REC; x represents the temperature of cold treatment.

### 2.3.2. Photosynthetic Parameters

Photosynthetic characteristics were measured using portable photosynthesis system (Li-6400xt, LI-COR, USA) from 9:00 to 12:00 by measuring the center leaf of the second or third healthy functional leaf from the top of the tested plant [26]. The instrument was set with a flow rate of 500 µmol s$^{-1}$, leaf chamber temperature and humidity were set at 25 °C and 65%, and the reference chamber $CO_2$ concentration was set at 400 µmol mol$^{-1}$. The PAR gradient was set as 0, 20, 50, 100, 150, 200, 400, 600, 800, 1000, 1200, 1400, 1600, and 1800 µmol m$^{-2}$ s$^{-1}$. Photosynthetic parameters were recorded by Li-6400xt automatically, including net photosynthetic rate ($P_n$), stomatal conductance ($G_s$), intercellular $CO_2$ concentration ($C_i$), and atmospheric $CO_2$ concentration ($C_a$) and others. Water-use efficiency (WUE) and stomatal limitation ($L_s$) were calculated based on Equations (4) and (5), respectively [27]. The maximum net photosynthetic rate ($Pn_{max}$) and apparent quantum yield (AQE) were obtained by fitting the light response curve using photosynthetic model [28].

$$WUE = P_n/T_r \tag{4}$$

$$L_s = 1 - C_i/C_a \tag{5}$$

In the equation, $T_r$ represents the transpiration rate.

### 2.3.3. Transient Chlorophyll Fluorescence Induction Kinetics

Plant Efficiency Analyzer (Pocket PEA, Hansatech, UK) was used to measure the transient chlorophyll fluorescence induction kinetics curves (OJIP transient) of strawberry leaves from 9:00 to 11:00. The saturating pulse intensity and actinic light intensity were set at 3000 and 200 µmol s$^{-1}$. The saturation pulse intensity was set at 3000 µmol s$^{-1}$ with a duration of 1s. The actinic light intensity was set at 200 µmol s$^{-1}$ with a duration of 9s. Each single-point measurement lasted for 210 s, with alternating saturation pulse and actinic light every 10 s in each cycle. We selected the central leaflet of the 2nd, 3rd, and 4th healthy functional leaves, from top to bottom, of each plant for measurement after 30 min's dark adaptation. The measuring points of one leaf were located on both sides of the midrib using clips, and the other two measuring one side of the midrib.

The OJIP transient contains abundant information that reflects the photochemical reaction state of the PSII reaction center.

The relative standardization of OJIP transient ($\Delta W$) represents the difference between two phases on the OJIP transient and CK. The calculation methods of $\Delta W$ for the O-K and O-J phases are shown in Equations (6) and (7). In this study, CK refers to 20 °C.

$$\Delta W_{OK} = [(F_t - F_0)/(F_k - F_o)]_{treatment} - [(F_t - F_0)/(F_k - F_o)]_{CK} \tag{6}$$

$$\Delta W_{OJ} = [(F_t - F_0)/(F_j - F_o)]_{treatment} - [(F_t - F_0)/(F_j - F_o)]_{CK} \tag{7}$$

In the equation, $\Delta W_{OK}$ and $\Delta W_{OJ}$ represent $\Delta W$ on O-K and O-J phase; $F_t$ and $F_0$ represent the fluorescence value (F) at one certain moment and minimum F; $F_j$ and $F_k$ represent the F at the J and K phase.

JIP-test [28] based on the energy flow through the photosynthetic membrane to analyze the measured chlorophyll fluorescence data. The JIP-test indices and terminology mentioned in this paper are presented in Table 1 [16,29].

**Table 1.** JIP-test indices and terminology used in the study.

| Terms and Formulas | Illustrations |
|---|---|
| | Specific energy fluxes (per active PSII reaction center): |
| ABS/RC | Absorption flux per RC |
| $TR_0$/RC | Trapped energy flux (leading to $Q_A$ reduction) per RC |
| $ET_0$/RC | Electron transport flux (further than $Q_A^-$) per RC |
| $RE_0$/RC | Electron flux reducing end electron acceptors at the PSI acceptor side per RC |
| $DI_0$/RC | Dissipated energy flux per RC |
| | Quantum yields and flux ratios: |
| $\varphi_{P_0} = TR_0/ABS$ | Maximum quantum yield of primary photochemistry ($F_v/F_m$) |
| $\psi_0 = ET_0/TR_0$ | Probability that a trapped exciton moves an electron into the electron transport chain beyond $Q_A^-$ |
| $\varphi_{E_0} = ET_0/ABS$ | Quantum yield of electron transport from $Q_A$ to $Q_B$ |
| $\varphi_{R_0} = RE_0/ABS$ | Quantum yield for reduction in end electron acceptors at the PSI acceptor side |
| $\varphi_{D_0} = 1 - \varphi_{P_0}$ | Quantum yield of heat dissipation |
| | Performance indexes: |
| $PI_{abs}$ | Performance index on absorption basis |

### 2.3.4. Solar-Induced Chlorophyll Fluorescence

Fluorescence radiance spectra were measured using fiber optic spectrometer (QE65 Pro, Ocean Optics, USA) from 11:00 to 12:00. The spectral resolution of the spectrometer is 1.4 nm, and spectral range from 645 to 800 nm. A 45° limiting device was used to fix the angle between the sun and the fiber optic probe. Whiteboard calibration was performed prior to data collection, radiance of whiteboard and leaves were sequentially recorded, consistent with the observation site for OJIP transient observation. Each measurement was repeated twice.

We adopted the absolute SIF (aFSR) algorithm proposed by Zhao et al. [30] to reconstruct the solar-induced chlorophyll fluorescence (SIF) spectrum. aFSR involves collecting the chlorophyll fluorescence radiation within the absorption band and using singular value decomposition (SVD) to obtain the principal components of the complete and continuous SIF spectrum based on the fluorescence radiance simulated by the SCOPE [31]. Compared with other SIF inversion algorithms such as FLD [32,33] and SFM [34], aFSR not only reflects the bimodal characteristics of SIF but also reconstructs the continuous SIF spectrum. The SIF information reconstructed by aFSR is absolute SIF (aSIF), which includes information from solar irradiance. To mitigate the unstable influence of the irradiance, the study also calculated relative SIF (rSIF), as shown in Equation (8).

$$rSIF = aSIF/E \tag{8}$$

In the equation, E represents the solar irradiance.

## 2.4. Statistical and Analytical Methods

IBM SPSS 24 (SPSS, IL, USA) was utilized for one-way ANOVA, Duncan's multiple comparisons ($p = 0.05$), and principal component analysis (PCA). Experimental data were presented in 'mean $\pm$ standard deviation (SD)' format. PCA is a core method for building minimal data sets. It reduces the dimensionality of data and extracts the main features, transforming multiple indicators into a few indicators. This method reduces the correlation between different indicators and makes them independent of each other [35]. Mean-centered standardized distance (MCSD) was one of the methods used in this study to extract diagnostic indicators of low-temperature stress in strawberry seedlings, with its calculation method shown in Equation (9).

$$D = \frac{|\mu_1 - \mu_2|}{\sigma_1 + \sigma_2} \tag{9}$$

In the equation, D represents the MCSD between two density functions; $\mu_1$ and $\mu_2$ represent the means of one parameter under two levels of low-temperature stress; $\sigma_1$ and $\sigma_2$ represent the SD of one parameter under two levels of low-temperature stress.

The model evaluation indicators used in this study were coefficient of determination ($R^2$), root-mean-square error (RMSE), and accuracy (Acc), as shown in Equations (10)–(12).

$$R^2 = \frac{\sum_i^n \left(\hat{y_i} - \overline{y}\right)^2}{\sum_i^n \left(y_i - \overline{y}\right)^2} \tag{10}$$

$$RMSE = \sqrt{\frac{\sum_i^n \left(\hat{y_i} - y_i\right)^2}{n}} \tag{11}$$

$$Acc = \frac{\hat{n_{true}}}{n} \tag{12}$$

In the equation, $\hat{y_i}$ represents the predicted value; $y_i$ represents the actual value; $\overline{y}$ represents the mean value; n represents the number of validation samples; $\hat{n_{true}}$ represents the number of correctly predicted validation samples.

## 2.5. Stress Diagnosis Comparative Models

Three types of machine learning models, multi-layer perceptron (MLP) [36], random forest (RF) [37], and support vector machine (SVM) [38], as well as multiple linear regression (MLR) [36] were used to evaluate the performance differences in different input data for strawberry seedlings low-temperature stress diagnosis applications. In this study, MLP was configured with 5 hidden layers, including 2 layers of 12 neurons, 2 layers of 24 neurons, and 1 layer of 6 neurons. The Relu function was used for activation, and the dropout layer was added before the fully connected layer to prevent overfitting. The model was trained for 3000 iterations. The number of trees in the RF was set to 100. The SVM model used the Linear kernel function with a Gamma value of 0.25.

## 3. Results

### 3.1. Effect of Low Temperature on Cell Membrane Permeability in Short-Day-Type and Long-Day-Type Strawberry Seedlings

The dynamic changes in relative electrical conductivity (REC) of the four strawberry cultivars under different levels of low-temperature stress are shown in Figure 2. Compared with 20 °C, there was little change in REC at 5 °C and 10 °C, indicating that the cell membrane of strawberry would not be damaged when the temperature was not lower than 5 °C. The most significant increase in REC occurred between 0 and −10 °C. Under

−15 °C to −20 °C, the extent of REC increase in different strawberry varieties significantly decreases and tends to the lowest level.

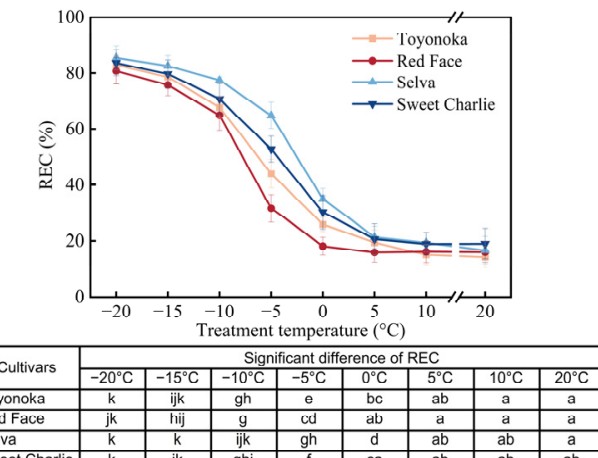

| Cultivars | Significant difference of REC | | | | | | | |
|---|---|---|---|---|---|---|---|---|
| | −20°C | −15°C | −10°C | −5°C | 0°C | 5°C | 10°C | 20°C |
| Toyonoka | k | ijk | gh | e | bc | ab | a | a |
| Red Face | jk | hij | g | cd | ab | a | a | a |
| Selva | k | k | ijk | gh | d | ab | ab | a |
| Sweet Charlie | k | jk | ghi | f | ca | ab | ab | ab |

**Figure 2.** Relative electrical conductivity (REC) of strawberry seedlings with different photoperiod types at different temperatures. Each value represents 'mean ± standard deviation (SD)'. Different lowercase letters indicate significant differences among treatments at the *p* < 0.05 level by Duncan's test. The color of the lines corresponds to the color of strawberry cultivars. 'Toyonoka' and 'Red Face' are short-day-type cultivars; 'Selva' and 'Sweet Charlie' are long-day-type cultivars.

Except for the commons mentioned above, there are differences in the characteristics of membrane permeability of strawberries with different photoperiods as temperature changes. At 0 °C treatment, the REC of the short-day-type strawberry 'Red Face' was only 2.08% higher than 20 °C. Both short-day-type cultivar 'Toyonoka' and long-day-type variety 'Sweet Charlie' showed a similar increase in REC, which was 11.62% and 11.18%, respectively. The REC of long-day-type strawberry 'Selva' increased by 18.3% compared to 20 °C. Based on the above, the membrane permeability of 'Red Face' was almost unaffected by the low temperature at 0 °C, while 'Toyonoka' and 'Sweet Charlie' showed slight influence, and 'Selva' was significantly affected. Between 0 and −5 °C, the extent of REC increase in different strawberry varieties ranked in the following order: 'Red Face' (13.46%), 'Toyonoka' (18.4%), 'Sweet Charlie' (22.72%), and 'Selva' (29.9%). At −5 °C, the smallest difference in REC was found between 'Toyonoka' and 'Sweet Charlie', with a difference of 8.77%. The REC of 'Red Face' was 21.29% lower than that of 'Sweet Charlie' and 36.01% lower than that of 'Selva'. This indicates that the REC of short-day-type strawberries was significantly lower than that of long-day-type strawberries. Between −5 and −10 °C, the REC of long-day-type strawberries was consistently higher than that of short-day-type varieties, but the rate of REC increase was opposite to that of the 0 °C to −5 °C. This suggests that the critical temperature at which short-day-type strawberries are damaged by low temperatures is lower than that of long-day-type varieties. At −10 °C, the difference in REC between the four strawberry varieties decreased significantly. The difference between 'Red Face' and 'Selva' decreased to 12.87%, indicating that the cellular membrane of the leaves of different photoperiod types of strawberries had been critically damaged at this temperature.

Using the Logistic equation to fit the REC curves of four strawberry cultivars, the fitted equations for REC with continuous temperature changes and semi-lethal temperature (LT$_{50}$) values were obtained, as shown in Table 2. All equations have determination coefficients ($R^2$) higher than 0.85, indicating a highly significant fit. Among the strawberry cultivars studied in this paper, the order of cold resistance from strong to weak is 'Red Face', 'Toyonoka', 'Sweet Charlie', and 'Selva', with LT$_{50}$ values of 7.67, −5.48, −3.83, and −2.13 °C, respectively. The LT$_{50}$ of the two short-day-type strawberry cultivars is lower

than those of the two long-day-type cultivars, indicating that short-day-type cultivars have better cold resistance during the seedling stage than long-day-type cultivars.

**Table 2.** REC fitting equations and $LT_{50}$ for four strawberry cultivars.

| Varieties | Logistic Equation | $R^2$ | $LT_{50}$ (°C) |
|---|---|---|---|
| Toyonoka | $y = 100/(1 \times 1.8272 \times \exp(0.1099 \times x))$ | 0.952 ** | −5.48 |
| Red Face | $y = 100/(1 \times 2.1750 \times \exp(0.1013 \times x))$ | 0.87 ** | −7.67 |
| Selva | $y = 100/(1 \times 1.2732 \times \exp(0.1133 \times x))$ | 0.94 ** | −2.13 |
| Sweet Charlie | $y = 100/(1 \times 1.4849 \times \exp(0.1033 \times x))$ | 0.938 ** | −3.83 |

Note: ** indicates significance of difference at 0.01 level. $LT_{50}$: Semi-lethal temperature.

### 3.2. Effect of Low Temperature on Photosynthetic Parameters in Short-Day-Type and Long-Day-Type Strawberry Seedlings

As the temperature decreased from 20 to −5 °C or lower, the stomata of leaves in four strawberry cultivars were observed to be almost completely closed (Figure 3a). At 5 °C, stomatal conductance (Gs) of 'Toyonoka' and 'Red Face' decreased by 44.28% and 39.76% compared to 20 °C, while 'Selva' and 'Sweet Charlie' showed a reduction of 52.04% and 44.98%, respectively. At 0 °C, the average Gs of short-day-type cultivars was 0.09, which was higher than that of long-day-type cultivars at 0.07.

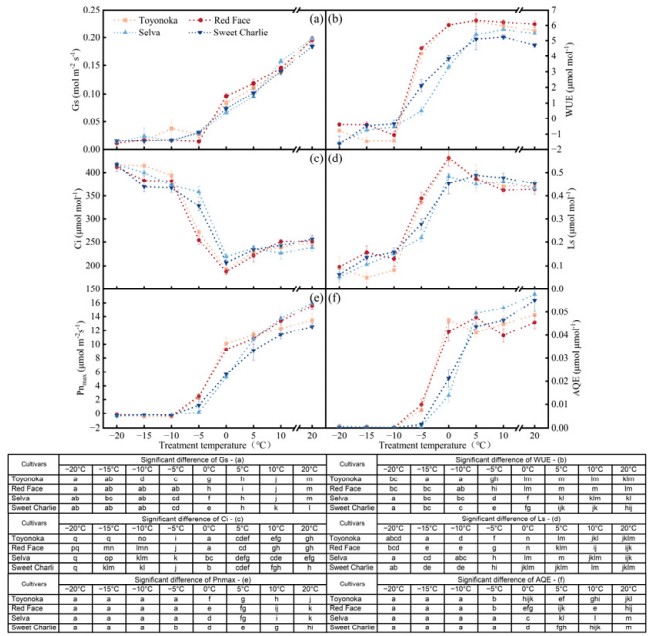

**Figure 3.** Photosynthetic parameters of strawberries seedlings with different photoperiod types at different temperatures: (**a**) Gs: Stomatal conductance; (**b**) WUE: Water-use efficiency; (**c**) Ci: Intercellular carbon dioxide concentration; (**d**) Ls: Stomatal limitation value; (**e**) $Pn_{max}$: Maximum net photosynthetic rate; (**f**) AQE: Apparent quantum efficiency. Each value represents 'mean ± standard deviation (SD)'. Different lowercase letters indicate significant differences among treatments at the $p < 0.05$ level by Duncan's test. The color of the lines corresponds to the color of strawberry cultivars. 'Toyonoka' and 'Red Face' are short-day-type cultivars; 'Selva' and 'Sweet Charlie' are long-day-type cultivars.

At −10 °C and below, the photosynthetic apparatus of the leaves was severely damaged by the low temperature, and respiration became dominant, resulting in a net photosynthetic rate of less than zero (Figure 3b). At −5, 0, and 5 °C, there were significant differences in water-use efficiency (WUE) between the short-day-type and long-day-type strawberry cultivars. The WUE of the short-day-type strawberry cultivars began to decrease at 0 to −5 °C with a decline of 1.71 µmol mol$^{-1}$, and then significantly decreased from −5 to −10 °C with a decrease of 5.62 µmol mol$^{-1}$. Long-day-type strawberries' WUE began to

decrease from 5 to 0 °C and continued until reaching the lowest level at −10 °C. 'Selva' showed a decrease in WUE of 2.14, 2.83, and 1.05 µmol mol$^{-1}$ in the temperature ranges of 5~0 °C, 0 to −5 °C and −5 to −10 °C, respectively, and 'Sweet Charlie' decreased in WUE of 1.25, 1.77, and 2.45 µmol mol$^{-1}$ in the same temperature ranges. It can be inferred that in the low-temperature range of 5 to −10 °C, the amount of assimilates produced by short-day-type strawberries per unit of water consumed is significantly higher than that of long-day-type cultivars. Among long-day-type varieties, 'Sweet Charlie' has a higher dry matter production capacity compared to 'Selva'.

In the temperature range from 20 to 0 °C, intercellular carbon dioxide concentration (Ci) (Figure 3c) of the four strawberry varieties continuously decreased with decreasing temperature, while the stomatal limitation value (Ls) (Figure 3d) showed a sustained increase except for 'Sweet Charlie'. It suggests that the decrease in the photosynthetic rate of strawberries between 15 and 0 °C is mainly due to stomatal factors. Between 0 and −20 °C, the trends of Ci and Ls changed opposite to those in 20 to 0 °C. Especially in the range of 0 to −10 °C, Ci significantly increased, gradually approaching the atmospheric $CO_2$ concentration, while Ls decreased significantly and tended to reach the lowest level. This indicates that the decline in photosynthesis of strawberries at temperatures below 0 °C is due to non-stomatal limiting factors. Additionally, Ci and Ls reflected the differences in photoperiodic types of strawberry varieties. In the range of 5~0 °C, the average decrease in Ci and increase in Ls in short-day-type cultivars were 35.26 and 1.18 µmol mol$^{-1}$, respectively, which were significantly higher than those of the long-day-type varieties, which were 21.72 and 0.0014 µmol mol$^{-1}$, respectively. At −5 °C, the mean Ci value of the short-day-type strawberries was 80.53 µmol mol$^{-1}$ lower than that of the long-day-types, while the mean Ls value was 0.13 µmol mol$^{-1}$ higher. At 0 °C, both short-day-type and long-day-type strawberries showed a turning point in the trends of Ci and Ls. The average Ci of short-day-type strawberries was 23.32 µmol mol$^{-1}$ higher than that of long-day-type varieties, while the average Ls of long-day-type strawberries was 0.09 µmol mol$^{-1}$ higher than that of short-day-type varieties.

Figure 3e,f demonstrate the changes in the maximum net photosynthetic rate (Pn$_{max}$) and apparent quantum efficiency (AQE) at different low-temperature gradients. At 0 °C, the Pn$_{max}$ of short-day-type strawberries was significantly higher than that of long-day-type strawberries, indicating that the inhibition of photosynthesis by low temperatures is greater in long-day-type strawberries than in short-day-type strawberries. The AQE also showed significant differences between different photoperiod strawberry types. In the range of 5 to 0 °C, the AQE of long-day-type strawberries began to decrease dramatically, with an average decrease of 62.42%, and approached the lowest level at −5 °C. For short-day-type strawberries, the significant decrease in AQE occurred in the temperature range of 0 to −5 °C, with an average decrease of 80.28%, and approached the lowest level at −10 °C treatment.

### 3.3. Effect of Low Temperature on Transient Chlorophyll Fluorescence Induction Kinetics in Short-Day-Type and Long-Day-Type Strawberry Seedlings

3.3.1. Effect of Low Temperature on OJIP Transient in Short-Day-Type and Long-Day-Type Strawberry Seedlings

The effect of low temperature on the transient chlorophyll fluorescence induction kinetics curve (OJIP transient) of four strawberry cultivars is shown in Figure 4. The OJIP transient can be divided into four phases: the O phase, representing the initial fluorescence; the J phase, representing the increase in fluorescence signal due to the accumulation of oxidized Q$_A$ and electron transfer beyond Q$_A$; the I phase, representing the fluorescence inflection point where Q$_A$ is converted to Q$_B$ and electrons transfer to PSI; and the P phase, representing the point of maximum fluorescence signal. Under treatments not lower than 0 °C, the OJIP transient of strawberry leaves in each variety exhibited distinct O, J, I, and P phases. At −5 °C, 'Toyonoka' and 'Red Face' showed weak P phase, while the P phase feature disappeared in 'Selva' and 'Sweet Charlie'. At lower temperatures, the characteristics of each phase in the OJIP curves disappeared and the chlorophyll fluorescence value (F) remained at a low level. It should be noted that under treatment at

−10 °C or lower, the J phase fluorescence is weak, with very little fluctuation over time, which exceeds the measurement accuracy of the instrument. This leads to an abnormal increase in the standardized curves of the O-K and O-J phases and the energy allocation parameters of the unit reaction center. Hence, the abnormal results obtained under −10 °C, −15 °C, and −20 °C treatments were not presented in the manuscript.

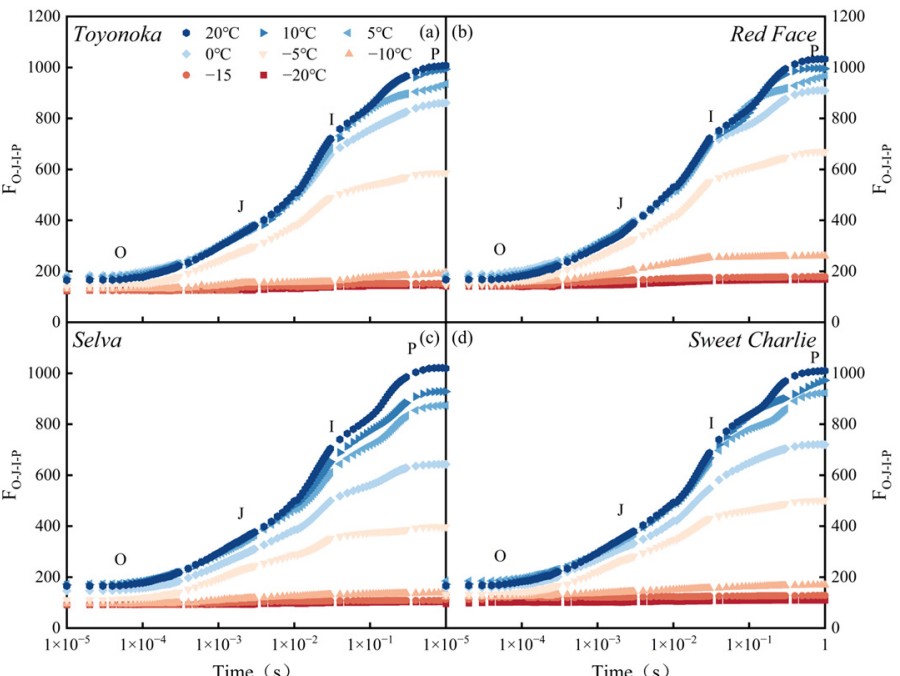

**Figure 4.** OJIP transient of strawberries seedlings with different photoperiod types at different temperatures: (**a**) OJIP transient of short-day-type 'Toyonoka'; (**b**) OJIP transient of short-day-type 'Red Face'; (**c**) OJIP transient of long-day-type 'Selva'; (**d**) OJIP transient of long-day-type 'Sweet Charlie'. Each curve represents the median level under that treatment. The color of the curve represents the temperature treatment. $F_{O-J-I-P}$: Fluorescence values (F) for each phase of O, J, I, and P.

Analyzing each phase, the initial fluorescence ($F_0$) of 'Toyonoka' and 'Selva' increased with decreasing temperature from 20 to 0 °C. The same phenomenon was observed in 'Selva' and 'Sweet Charlie' when the temperature was not lower than 5 °C. This indicates that the supply of electrons in the electron transfer chain decreases, leading to insufficient utilization of light energy. Subsequently, both short-day-type and long-day-type strawberries showed a decrease in $F_0$ with the decrease in treatment temperature, starting from −5 and 0 °C, respectively. It indicates a reduction in the efficiency of light energy conversion in photosynthesis, and a decline in physiological activity of the plants. At 0 °C, the differences in $F_j$ and $F_i$ of the short-day-type strawberries compared to those at 5 °C, 10 °C, and 20 °C are relatively small. However, for the long-day-type varieties, $F_j$ and $F_i$ under 0 °C treatment were significantly lower than those at 5 °C and above. The $F_i$ of Selva showed a significant decrease at 10 °C and 5 °C. At the P phase, the differential effects of low temperatures on the OJIP transient reach maximum. At 0 °C, the $F_p$ of 'Toyonoka' and 'Red Face' decreased by 14.58% and 12.00%, respectively, compared with 20 °C. The $F_p$ of 'Selva' and 'Sweet Charlie' showed a significant decrease of 36.90% and 28.71%, respectively, compared to 20 °C. At −5 °C, the $F_p$ values of 'Toyonoka' and Red Face were 58.13% and 64.76% of their respective CK maximum fluorescence intensities, while the $F_p$ values of 'Selva' and 'Sweet Charlie' decreased more than those of the short-day-type strawberries, by 38.96% and 49.7% at 20 °C, respectively.

The relative normalized OJIP transient $\Delta W_{ok}$ and $\Delta W_{oj}$ between O-K and O-J phases under temperatures between 10 and −5 °C are shown in Figure 5. Due to abnormally large values of $\Delta W$ recorded by the instrument under −10, −15, and −20 °C treatments, they are not presented in this sector.

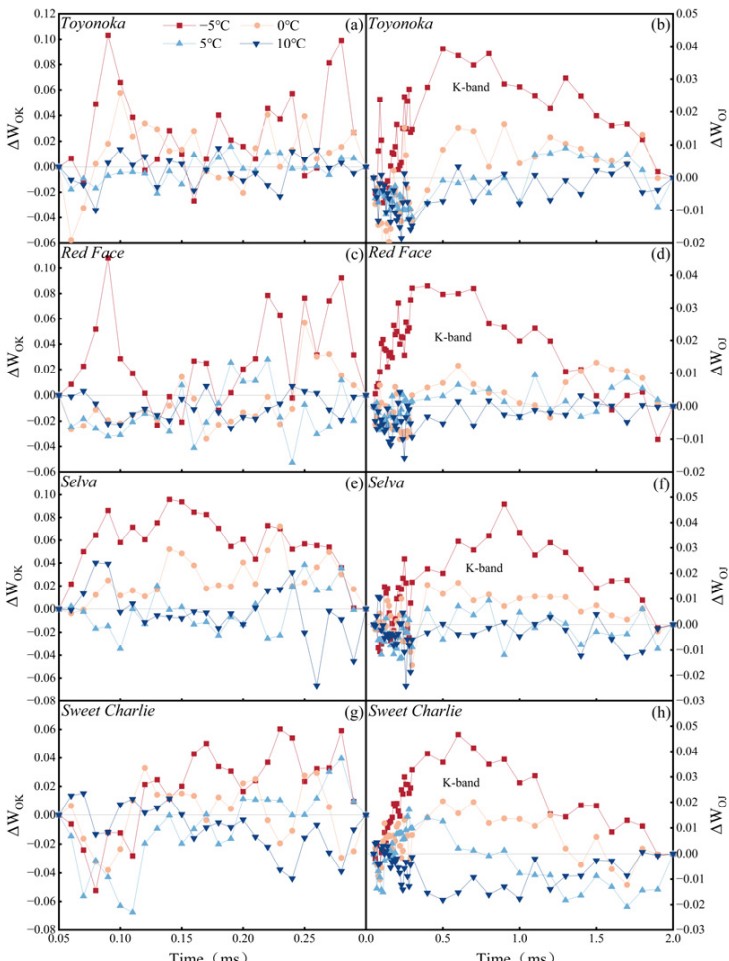

**Figure 5.** $\Delta W_{ok}$ and $\Delta W_{oj}$ of strawberries seedlings with different photoperiod types at different temperatures: (**a,b**) $\Delta W_{ok}$ and $\Delta W_{oj}$ of short-day-type 'Toyonoka'; (**c,d**) $\Delta W_{ok}$ and $\Delta W_{oj}$ of short-day-type 'Red Face'; (**e,f**) $\Delta W_{ok}$ and $\Delta W_{oj}$ of long-day-type 'Selva'; (**g,h**) $\Delta W_{ok}$ and $\Delta W_{oj}$ of long-day-type 'Sweet Charlie'. Each curve represents the median level under that treatment. The color of the curve represents the temperature treatment. $\Delta W_{ok}$: Relative standardization OJIP transient in O-K phase; $\Delta W_{oj}$: Relative standardization OJIP transient in O-J phase.

$\Delta W_{OK}$ reflects the continuity of energy transfer between the PSII reaction centers. When $\Delta W_{OK}$ is less than zero, it indicates the photosynthetic membrane system is intact and energy transfer is unobstructed. When it is greater than zero, it indicates the energy transfer is blocked, and the energy that is not effectively transferred will damage the stability of the PSII center [39]. At 5 °C and 10 °C, $\Delta W_{OK}$ of the four strawberry cultivars were distributed near the zero line, indicating that the energy transfer capacity of the PSII center between thylakoid membranes was similar to 20 °C and not yet affected by low-temperature stress at these two temperatures. At 0 °C, the $\Delta W_{OK}$ curve of 'Selva' mostly became positive, while 'Toyonoka' and 'Sweet Charlie' showed more positive values than at 5 °C and 10 °C, but still had a few negative points. On the other hand, for 'Red Face', more negative values remained at 0 °C. It can be observed that at 0 °C, the energy transfer between PSII center complexes in 'Selva' is affected, while 'Toyonoka' and 'Sweet Charlie' are slightly affected, and 'Red Face' is basically unaffected. At −5 °C, the energy transfer between the PSII centers of all strawberry varieties was affected, with 'Selva' being the most affected.

When the K-band is greater than zero, it indicates the inactivation of the oxygen-evolving complex (OEC), while a negative K-band indicates its activation. The degree of increase in $\Delta W_{OJ}$ reflects the extent of OEC damage [40]. At 5 °C and 10 °C, $\Delta W_{OJ}$ of different strawberry

varieties are distributed around zero. When the temperature drops to 0 °C, 'Toyonoka', 'Selva', and 'Sweet Charlie' all show significant K-bands, and Red Face also shows a K-band, but it is weaker compared to the other three varieties. Indicating that 'Red Face' has a stronger adaptability than the other three cultivars at 0 °C. At −5 °C, significant K-bands were observed in all four strawberry varieties between 0.5 and 1 s, showing the strongest impact on OEC. It is worth noting that at −5 °C, the K-band peak values for the long-day-type strawberry cultivars exceeded 0.04, while for the short-day-type strawberry varieties 'Toyonoka' was 0.039 and 'Red Face' was 0.037. This suggests that the short-day-type strawberries have stronger low-temperature adaptation of OEC at −5 °C compared to the long-day-type varieties.

### 3.3.2. Effect of Low Temperature on Activity of PSII Reaction Center in Short-Day-Type and Long-Day-Type Strawberry Seedlings

The changes in the PSII reaction center activity of four strawberry varieties under different degrees of low-temperature stress are shown in Table 3. The amount of light energy absorbed per reaction center (ABS/RC) and energy required to reduce $Q_A$ per reaction center ($TR_0/RC$) per reaction center exhibited an increasing trend followed by a decreasing trend as temperature decreased. The ABS/RC of 'Selva' and 'Sweet Charlie' peaked at 5 °C, while 'Toyonoka' reached maximum at 0 °C and 'Red Face 'at −5 °C. At the control temperature, the average ABS/RC was 1.475 for short-day-type strawberries and 1.423 for long-day-type varieties, respectively. At −5 °C, the order of ABS/RC from high to low for the four strawberry varieties was 'Red Face' (1.591), 'Toyonoka' (1.499), 'Sweet Charlie' (1.456), and 'Selva' (1.416). 'Toyonoka', Selva', and 'Sweet Charlie' reached the maximum $TR_0/RC$ at 10 °C, while 'Red Face' reached the peak at 5 °C. At temperatures of 5, 0, and −5 °C, the decrease in $TR_0/RC$ of long-day-type strawberries was greater than that of short-day-type cultivars. At −5 °C, the $TR_0/RC$ of 'Red Face', 'Toyonoka', 'Sweet Charlie', and 'Selva' decreased by 0.027, 0.087, 0.1, and 0.128, respectively, compared to 5 °C. The changes in the energy used for electron transfer per reaction center ($ET_0/RC$) were not significant at 5 °C and above. However, at 0 °C, the $ET_0/RC$ of all strawberry varieties started to decrease, with long-day-type strawberries showing a greater decrease than short-day-type cultivars. At −5 °C, the $ET_0/RC$ of long-day-type strawberries was significantly lower than that of short-day-type cultivars, with a decreasing order of 'Red Face' (0.12), 'Toyonoka' (0.161), 'Sweet Charlie' (0.245), and 'Selva' (0.306), compared to 20 °C. The level of the PSI acceptor side electron flux reduction ($RE_0/RC$) in each strawberry variety decreased with decreasing temperature, similar to the pattern of $ET_0/RC$. Specifically, in long-day-type strawberries, $RE_0/RC$ began to decrease significantly at 0 °C. For short-day-type strawberries, the decrease in $RE_0/RC$ was less pronounced than in long-day-type cultivars at 5~0 °C and 0 to −5 °C. The energy dissipation per reaction center ($DI_0/RC$) increases with decreasing temperature. Different photoperiod types of strawberry varieties are similar in $DI_0/RC$ from 20 to 0 °C at each low-temperature gradient. However, at −5 °C, $DI_0/RC$ of two long-day-type varieties increased rapidly, with a mean value of 0.383, significantly higher than 0.352 for the mean value of two short-day-type varieties.

**Table 3.** Activity of PSII reaction center of strawberries with different photoperiod types at different temperatures.

| Cultivars | Treatment Temperature (°C) | ABS/RC | $TR_0$/RC | $ET_0$/RC | $RE_0$/RC | $DI_0$/RC | $\psi_0$ | $\varphi P_0$ | $\varphi E_0$ | $\varphi R_0$ | $\varphi D_0$ |
|---|---|---|---|---|---|---|---|---|---|---|---|
| Toyonoka | 20 | 1.496 ± 0.052ghi | 1.202 ± 0.138efg | 0.953 ± 0.077ijk | 0.421 ± 0.017hij | 0.245 ± 0.013b | 0.79 ± 0.016jkl | 0.84 ± 0.009m | 0.657 ± 0.008kl | 0.285 ± 0.012hij | 0.16 ± 0.009a |
| | 10 | 1.496 ± 0.019ghi | 1.237 ± 0.014ghi | 0.971 ± 0.018kl | 0.416 ± 0.016hi | 0.259 ± 0.008c | 0.785 ± 0.008ijk | 0.827 ± 0.004jk | 0.649 ± 0.005jk | 0.278 ± 0.012ghi | 0.173 ± 0.004cd |
| | 5 | 1.518 ± 0.018i | 1.232 ± 0.015fghi | 0.959 ± 0.021jk | 0.409 ± 0.01h | 0.286 ± 0.007d | 0.778 ± 0.008hij | 0.812 ± 0.004h | 0.631 ± 0.007i | 0.27 ± 0.008g | 0.188 ± 0.004f |
| | 0 | 1.519 ± 0.022i | 1.198 ± 0.019ef | 0.908 ± 0.018fg | 0.358 ± 0.013fg | 0.321 ± 0.006e | 0.758 ± 0.005f | 0.788 ± 0.003f | 0.602 ± 0.005f | 0.236 ± 0.008f | 0.212 ± 0.003h |
| | −5 | 1.499 ± 0.025hi | 1.145 ± 0.016c | 0.792 ± 0.022c | 0.249 ± 0.017c | 0.354 ± 0.012f | 0.691 ± 0.016c | 0.764 ± 0.005c | 0.528 ± 0.014c | 0.166 ± 0.011c | 0.236 ± 0.005k |
| Red Face | 20 | 1.454 ± 0.083cdef | 1.253 ± 0.067ij | 1 ± 0.042m | 0.44 ± 0.019k | 0.234 ± 0.034ab | 0.798 ± 0.016l | 0.837 ± 0.02lm | 0.67 ± 0.005m | 0.297 ± 0.016k | 0.163 ± 0.02ab |
| | 10 | 1.518 ± 0.024i | 1.262 ± 0.017ij | 0.988 ± 0.013lm | 0.434 ± 0.013jk | 0.256 ± 0.012c | 0.782 ± 0.005hijk | 0.831 ± 0.006kl | 0.65 ± 0.003jkl | 0.286 ± 0.01ij | 0.169 ± 0.006bc |
| | 5 | 1.562 ± 0.017j | 1.275 ± 0.02j | 0.991 ± 0.026lm | 0.43 ± 0.017ijk | 0.287 ± 0.007d | 0.777 ± 0.009ghi | 0.816 ± 0.006hi | 0.634 ± 0.01i | 0.275 ± 0.01gh | 0.184 ± 0.006ef |
| | 0 | 1.581 ± 0.016j | 1.255 ± 0.022ij | 0.961 ± 0.02jk | 0.368 ± 0.019f | 0.327 ± 0.007e | 0.766 ± 0.005fg | 0.793 ± 0.006f | 0.609 ± 0.007g | 0.233 ± 0.012f | 0.207 ± 0.006h |
| | −5 | 1.591 ± 0.02j | 1.241 ± 0.017hij | 0.88 ± 0.017e | 0.28 ± 0.016d | 0.35 ± 0.008f | 0.709 ± 0.013d | 0.78 ± 0.004e | 0.553 ± 0.009d | 0.176 ± 0.011d | 0.22 ± 0.004i |
| Selva | 20 | 1.407 ± 0.094a | 1.144 ± 0.047c | 0.916 ± 0.02gh | 0.427 ± 0.015ijk | 0.227 ± 0.014a | 0.794 ± 0.018kl | 0.836 ± 0.008lm | 0.66 ± 0.012l | 0.308 ± 0.014l | 0.162 ± 0.004a |
| | 10 | 1.419 ± 0.011ab | 1.16 ± 0.015cd | 0.9 ± 0.015efg | 0.421 ± 0.018hij | 0.259 ± 0.005c | 0.776 ± 0.007ghi | 0.817 ± 0.005hi | 0.634 ± 0.007i | 0.297 ± 0.013k | 0.183 ± 0.005ef |
| | 5 | 1.436 ± 0.019abcf | 1.152 ± 0.012cd | 0.89 ± 0.015ef | 0.435 ± 0.017jk | 0.284 ± 0.008d | 0.772 ± 0.007g | 0.802 ± 0.004g | 0.619 ± 0.005h | 0.303 ± 0.011kl | 0.198 ± 0.004g |
| | 0 | 1.428 ± 0.025abc | 1.105 ± 0.018b | 0.798 ± 0.025c | 0.319 ± 0.016e | 0.324 ± 0.014e | 0.723 ± 0.013e | 0.773 ± 0.008d | 0.559 ± 0.012d | 0.223 ± 0.013e | 0.227 ± 0.008j |
| | −5 | 1.416 ± 0.021ab | 1.024 ± 0.021a | 0.61 ± 0.045a | 0.16 ± 0.014a | 0.391 ± 0.018h | 0.595 ± 0.032a | 0.724 ± 0.012a | 0.431 ± 0.028a | 0.113 ± 0.013a | 0.276 ± 0.012m |
| Sweet Charlie | 20 | 1.439 ± 0.016bcde | 1.203 ± 0.015efg | 0.943 ± 0.015ij | 0.46 ± 0.015l | 0.236 ± 0.005ab | 0.784 ± 0.005ijk | 0.836 ± 0.003lm | 0.655 ± 0.005jkl | 0.32 ± 0.011m | 0.164 ± 0.003ab |
| | 10 | 1.47 ± 0.021efgh | 1.209 ± 0.017efgh | 0.95 ± 0.013ijk | 0.459 ± 0.019l | 0.261 ± 0.006c | 0.786 ± 0.003ijk | 0.822 ± 0.003ij | 0.646 ± 0.004j | 0.312 ± 0.016lm | 0.178 ± 0.003dr |
| | 5 | 1.476 ± 0.019fgh | 1.181 ± 0.021de | 0.932 ± 0.017hi | 0.435 ± 0.015jk | 0.295 ± 0.008d | 0.789 ± 0.003jkl | 0.8 ± 0.006g | 0.632 ± 0.004i | 0.295 ± 0.011jk | 0.2 ± 0.006g |
| | 0 | 1.465 ± 0.021defg | 1.144 ± 0.01c | 0.834 ± 0.021d | 0.353 ± 0.015f | 0.321 ± 0.012e | 0.729 ± 0.012e | 0.781 ± 0.006e | 0.569 ± 0.01e | 0.241 ± 0.01f | 0.219 ± 0.006i |
| | −5 | 1.456 ± 0.015cdef | 1.081 ± 0.019b | 0.698 ± 0.031b | 0.209 ± 0.018b | 0.375 ± 0.014g | 0.645 ± 0.017b | 0.742 ± 0.009b | 0.479 ± 0.019b | 0.143 ± 0.013b | 0.258 ± 0.009l |

Note: Each value represents 'mean ± standard deviation (SD)'. Different lowercase letters indicate significant differences among treatments at the $p < 0.05$ level by Duncan's test. ABS/RC: Absorption flux per RC; $TR_0$/RC: Trapped energy flux (leading to $Q_A$ reduction) per RC; $ET_0$/RC: Electron transport flux (further than $Q_A^-$) per RC; $RE_0$/RC: Electron flux reducing end electron acceptors at the PSI acceptor side per RC; $DI_0$/RC: Dissipated energy flux per RC; $\psi_0$: Probability that a trapped exciton moves an electron into the electron transport chain beyond $Q_A^-$; $\varphi P_0$: Maximum quantum yield of primary photochemistry ($F_v/F_m$); $\varphi E_0$: Quantum yield of electron transport from $Q_A$ to $Q_B$; $\varphi R_0$: Quantum yield for reduction in end electron acceptors at the PSI acceptor side; $\varphi D_0$: Quantum yield of heat dissipation. 'Toyonoka' and 'Red Face' are short-day-type cultivars; 'Selva' and 'Sweet Charlie' are long-day-type cultivars.

At 0 °C, the probability of electron transfer from the captured excitons to other electron acceptors beyond $Q_A^-$ ($\psi_0$) by long-day-type strawberries reduced significantly, with an average decrease of 6.3% compared to 20 °C. The $\psi_0$ of short-day-type strawberries is 1.99% lower than 20 °C. At −5 °C, the $\psi_0$ of short-day-type cultivars decreases significantly, with 'Toyonoka', 'Red Face', 'Selva', and 'Sweet Charlie' showing reductions of 0.099, 0.128, 0.199, and 0.139, respectively, compared to 20 °C. The maximum PSII quantum efficiency ($\varphi P_0$) of the four strawberry cultivars under 20 °C treatment is comparable. $\varphi P_0$ decreases with decreasing temperature, and $\varphi P_0$ is lower than that of 20 °C under different temperature treatments. The rate of decrease in $\varphi P_0$ of long-day-type strawberries is greater than that of short-day-type cultivars under low-temperature stress. Different strawberry species show $\varphi P_0$ over 0.8 in treatments at 5 °C and above, indicating that strawberries are still healthy when the temperature is not lower than 5 °C. At 0 °C, the $\varphi P_0$ of the four varieties decreased to below 0.8. At −5 °C, the decrease in $\varphi P_0$ of long-day-type strawberries is greater than that of short-day-type cultivars again, with 'Toyonoka' and 'Red Face' having $\varphi P_0$ of 0.764 and 0.78, respectively. While 'Selva' and 'Sweet Charlie' have $\varphi P_0$ of 0.724 and 0.742. This suggests that the photosynthesis of long-day-type strawberries is more susceptible to low temperature. The quantum yield of electron transfer ($\varphi E_0$) for short-day-type and long-day-type strawberries decreases with decreasing temperature. It slowly decreases and shows a similar trend between 20 °C, 10 °C, and 5 °C treatments. At 0 °C, the $\varphi E_0$ of 'Toyonoka' and 'Red Face' is higher than 0.6, while 'Selva' and 'Sweet Charlie' have $\varphi E_0$ about 0.56 and 0.57, respectively. At −5 °C, the difference in $\varphi E_0$ between different photoperiod types of strawberries further increases. At this temperature, compared to 20 °C, 'Selva' decreases by 34.7%, 'Sweet Charlie' decreases by 26.87%, 'Toyonoka' and 'Red Face' decreases by 19.63% and 17.46%, respectively. The quantum yield of the terminal electron acceptors of PSI receptor ($\varphi R_0$) in long-day-type strawberries is higher than that in short-day-type cultivars at 5 °C, 10 °C, and 20 °C. The $\varphi R_0$ of each strawberry variety decreases with decreasing temperature, and the decrease in $\varphi R_0$ of long-day-type strawberries is greater than that of short-day-type varieties. At 0 °C, $\varphi R_0$ of different photoperiod types of strawberries are similar. At −5 °C, $\varphi R_0$ of two long-day-type strawberry species are lower than short-day-type varieties. At this temperature, the values of $\varphi R_0$ for 'Toyonoka', 'Red Face', 'Selva', and 'Sweet Charlie' were 58.25%, 59.26%, 36.69%, and 44.69% of CK, respectively. When exposed to low-temperature stress, unused light energy captured by plants increases, requiring more thermal dissipation to maintain heat balance. Therefore, within the temperature range of 10 to −5 °C, the quantum yield for thermal dissipation ($\varphi D_0$) of different strawberry varieties increases with decreasing temperature. $\varphi D_0$ of long-day-type strawberries is consistently higher than short-day-type cultivars, and the difference in $\varphi D_0$ between different photoperiod types of strawberries increases as the temperature decreases.

### 3.3.3. Effect of Low Temperature on Performance Index Based on Absorption in Short-Day-Type and Long-Day-Type Strawberry Seedlings

Performance index on absorption basis ($PI_{abs}$) is an absorption-based performance index that can comprehensively evaluate the damage of the photosynthetic apparatus under low-temperature stress. As shown in Figure 6, different strawberry varieties showed a similar degree of decline from 20 to 5 °C. Between 5 and −5 °C, the decline rates of long-day-type cultivars 'Selva' and 'Sweet Charlie' was higher than those of the short-day-type varieties 'Toyonoka' and 'Red Face'. At 0 °C, the $PI_{abs}$ of 'Toyonoka' and 'Red Face' decreased by 39.41% and 43.05%, respectively, compared to 20 °C. Meanwhile, 'Selva' and 'Sweet' Charlie exhibited a reduction of 56.13% and 49.61%, respectively. At −5 °C, the $PI_{abs}$ of 'Selva' and 'Sweet Charlie' was only 18.74% and 26.99% of their 20 °C, while 'Toyonoka' and 'Red Face' were 37.27% and 38.08% of the 20 °C, respectively. The above indicates that short-day-type strawberries have a higher light absorption and utilization efficiency in PSII than long-day-type varieties at 0 °C and −5 °C. As the temperature dropped to −10 °C,

the $PI_{abs}$ of all tested strawberry varieties approached zero. The plants were no longer able to effectively convert light energy into biological energy.

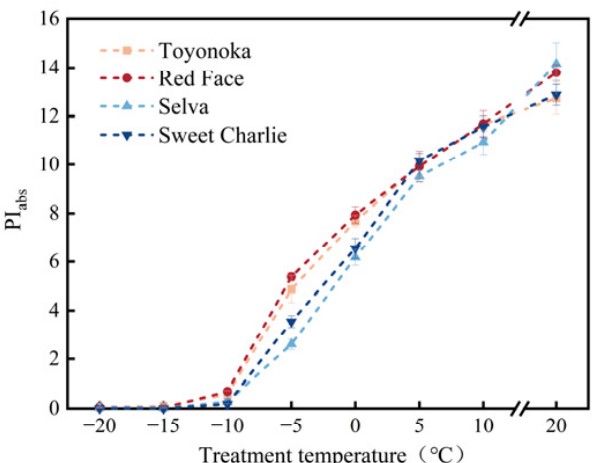

| Cultivars | Significant difference of Plabs | | | | | | | |
|---|---|---|---|---|---|---|---|---|
| | −20°C | −15°C | −10°C | −5°C | 0°C | 5°C | 10°C | 20°C |
| Toyonoka | a | a | b | e | i | k | m | n |
| Red Face | a | a | b | f | i | k | m | o |
| Selva | a | a | a | c | g | j | l | p |
| Sweet Charlie | a | a | a | d | h | k | m | n |

**Figure 6.** $PI_{abs}$ of strawberries seedlings with different photoperiod types at different temperatures. Each value represents 'mean ± standard deviation (SD)'. Different lowercase letters indicate significant differences among treatments at the $p < 0.05$ level by Duncan's test. The color of the letters corresponds to the color of strawberry cultivars. $PI_{abs}$: Absorption-based performance index. 'Toyonoka' and 'Red Face' are short-day-type cultivars; 'Selva' and 'Sweet Charlie' are long-day-type cultivars.

*3.4. Effect of Low Temperature on Solar-Induced Chlorophyll Fluorescence in Short-Day-Type and Long-Day-Type Strawberry Seedlings*

The changes in leaf solar-induced chlorophyll fluorescence (SIF) of four tested strawberry varieties under different levels of low-temperature stress are shown in Figure 7. The main features of absolute SIF (aSIF) are reflected in two peaks near 685 nm and 740 nm, and the aSIF at 740 nm changes more dramatically under low-temperature stress than at 685 nm, indicating a stronger sensitivity to low temperature. Compared to aSIF, relative SIF (rSIF) highlights SIF signals within the $O_2$-B and $O_2$-A absorption bands near 685 nm and 760 nm, respectively. The difference in SIF characteristics between strawberry varieties with different photoperiod types is most evident at 0 °C. After 0 °C treatment, aSIF at 742 nm ($aSIF_{742}$) and rSIF at 760 nm ($rSIF_{760}$) of short-day-type strawberries are significantly higher than long-day-type varieties. $aSIF_{742}$ and $rSIF_{760}$ of 'Toyonoka', 'Red Face', 'Selva', and 'Sweet Charlie' were 1.13, 1.117, 0.735, and 0.906 and 0.125, 0.122, 0.096, and 0.110, respectively, accounting for 67.19%, 69.95%, 43.52%, and 55.62% and 71.23%, 72.8%, 52.40%, and 63.56% of their respective CK. Therefore, SIF can reveal the differences in low-temperature stress between short-day-type and long-day-type strawberries. The study selected aSIF at 683 and 742 nm ($aSIF_{683}$ and $aSIF_{742}$) and rSIF at 687, 742, 760, and 763 nm ($rSIF_{687}$, $rSIF_{742}$, $rSIF_{760}$, and $rSIF_{763}$) as SIF features to support the establishment of low-temperature stress diagnosis model for strawberries.

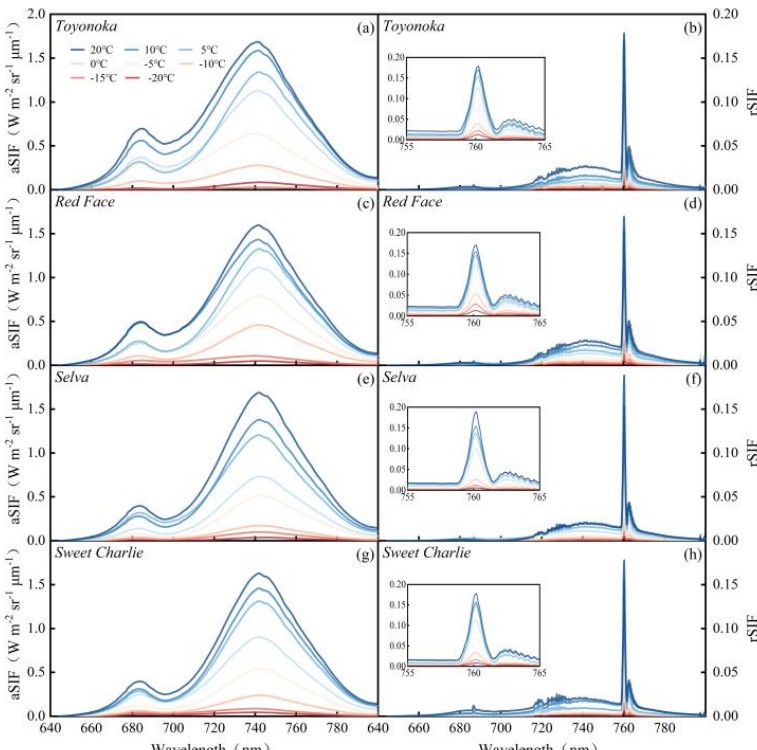

**Figure 7.** SIF spectrum of strawberries seedlings with different photoperiod types at different temperatures: (**a**,**b**) aSIF and rSIF of short-day-type 'Toyonoka'; (**c**,**d**) aSIF and rSIF of short-day-type 'Red Face'; (**e**,**f**) aSIF and rSIF of long-day-type 'Selva'; (**g**,**h**) aSIF and rSIF of long-day-type 'Sweet Charlie'. The windows in (**b**,**d**,**f**,**h**), are enlargements of the rSIF feature at 760 nm. Each curve represents the median level under that treatment. The color of the curve represents the temperature treatment. aSIF: absolute SIF; rSIF: relative SIF.

*3.5. Construction of Non-Destructive Diagnosis Models for Low-Temperature Stress in Strawberry Seedlings*

3.5.1. Grading of Low-Temperature Stress Levels in Strawberry Seedlings

To better illustrate the extent of damage caused by low temperatures in strawberries, the study comprehensively established a grading of low-temperature stress levels in strawberry seedlings (Table 4) by combining the REC and photosynthetic fluorescence indicators under different cold treatments mentioned earlier. We defined Level 0 as no cold stress, and Level 1 as mild stress where electrolytes began to leak from leaves and where plants could still recover if timely measures were taken. When more than half of the electrolytes had leaked from leaves and the physiological activities were severely affected, the plants were defined as Level 2, in a semi-dying state. Level 3 indicates the loss of physiological activities in plants, as shown by various observation indices, and this mainly occurs under temperature treatments at $-10\,^\circ\text{C}$ or lower.

**Table 4.** Grade of low-temperature stress levels in strawberry seedlings.

| Relative Electrical Conductivity (%) | Low-Temperature Stress Level |
|---|---|
| $0 \leq \text{REC} \leq 30$ | Level 0 |
| $30 < \text{REC} \leq 50$ | Level 1 |
| $50 < \text{REC} \leq 65$ | Level 2 |
| $65 < \text{REC} \leq 100$ | Level 3 |

Note: REC: Relative electrical conductivity.

### 3.5.2. Optimization of Non-Destructive Diagnostic Indicators for Low-Temperature Stress in Strawberry Seedlings

Our research found that parameters of photosynthesis, transient chlorophyll fluorescence induction kinetics, and SIF intensity could all reflect changes in the photosynthetic physiological activity of strawberry seedlings under low-temperature stress. However, the intrinsic plant information reflected by different types of indicators varies. Thus, this study evaluated the diagnostic performance of different types of observation indicators and different combinations of indicators for low-temperature stress in strawberry seedlings. We utilized three types of photosynthetic physiological parameters, including six photosynthesis parameters (Gs, WUE, Ci, Ls, $Pn_{max}$, AQE), 11 chlorophyll fluorescence parameters (ABS/RC, $TR_0$/RC, $ET_0$/RC, $DI_0$/RC, $RE_0$/RC, $\psi_0$, $\varphi P_0$, $\varphi E_0$, $\varphi D_0$, $\varphi R_0$, $PI_{abs}$), and six SIF parameters ($aSIF_{683}$, $aSIF_{742}$, $rSIF_{687}$, $rSIF_{742}$, $rSIF_{760}$, $rSIF_{763}$), as three sets of input data for diagnose models. The study also extracted two additional sets of data using principal component analysis (PCA) and mean-centered standardized distance (MCSD) methods. The two most significant principal components (PCs) from each type of data were extracted using PCA. Two distance characteristic parameters (DCPs) with optimal distinguishability between adjacent stress levels were selected from each type of data using MCSD. The selected PCs and DCPs are shown in Table 5.

**Table 5.** PCs and DCPs selected based on different types of observation indicators.

| PCA | PCs | WUE | $Pn_{max}$ | $\varphi P_0$ | $RE_0$/RC | $aSIF_{742}$ | $rSIF_{760}$ |
|---|---|---|---|---|---|---|---|
| | Loadings | 0.969 | 0.97 | 0.931 * | 0.875 ** | 0.985 | 0.989 |
| | DCPs | Ci | AQE | $\varphi R_0$ | $PI_{abs}$ | $aSIF_{742}$ | $rSIF_{760}$ |
| MCSD | Level 0&1 | 0.99 | 1.33 | 2.64 | 1.88 | 2.35 | 2.49 |
| | Level 0&2 | 4.01 | 3.04 | 3.61 | 2.85 | 3.72 | 4.24 |
| | Level 0&3 | 3.82 | 3.95 | 1.9 | 3.61 | 3.06 | 3.43 |
| | Level 1&2 | 4.33 | 2.23 | 1.09 | 1.5 | 1.97 | 1.72 |
| | Level 1&3 | 3.9 | 3.6 | 1.02 | 3.42 | 2.09 | 2.24 |
| | Level 2&3 | 2.86 | 2.18 | 0.82 | 3.42 | 2.09 | 2.42 |

Note: * represents the factor with the highest contribution in the PC1; ** represents the factor with the highest contribution in the PC2. PCs: Principal components extracted based on principal component analysis (PCA), including WUE, $Pn_{max}$, $\varphi P_0$, $RE_0$/RC, $aSIF_{742}$, and $rSIF_{760}$; DCPs: Distance characteristic parameters extracted based on mean-centered standardized distance (MCSD), including Ci, AQE, $\varphi R_0$, $PI_{abs}$, $aSIF_{742}$, and $rSIF_{760}$.

To evaluate the diagnostic accuracy of the five sets of input data for assessing the severity of low-temperature stress in strawberry seedlings, the study utilized three classical machine learning algorithms, MLP, RF, and SVM, as well as MLR. These models were trained and tested on the observation data to simulate practical applications for diagnosing low-temperature stress in strawberry seedlings. Due to limitations in observation, the amount of data for photosynthesis parameters was the smallest among the three types of data. To ensure fairness in testing, the number of samples in each type of data was equal to the number of photosynthesis parameters. A total of 96 valid data sets were obtained, with 72 used as training samples and 24 used for accuracy testing. The diagnostic accuracy for each low-temperature stress level based on different input parameters and models is shown in Table 6. The accuracy and error of diagnosing low-temperature stress levels using DCPs as input parameters are significantly better than those of other types of input parameters in different models. The diagnostic accuracy based on PCs was 95%, ranking second, showing lower stability than DCPs. The overall accuracy based on a single type of photosynthetic physiological parameter was all below 90%, indicating that the information carried by a single type of parameter is insufficient to describe the overall stress condition of plants. The combination of different types of indicators can more comprehensively describe the degree of low-temperature stress that plants experience. In addition, we found that SVM may not be suitable for diagnosing low-temperature stress. When using photosynthetic parameters, SIF parameters, and PCs as input parameters, the diagnostic accuracy was distributed at around 80%. Even when DCPs were input into the model, the

recognition accuracy did not reach 90% either. However, when chlorophyll fluorescence parameters were used as input parameters, the accuracy improved, which may be due to the larger amount of chlorophyll fluorescence parameters used as input. In summary, DCPs demonstrate a superior diagnostic ability for assessing the severity of low-temperature stress in strawberry seedlings compared to single-type photosynthetic parameters and PCs. The diagnostic model established based on MLR is presented in Table 7.

**Table 6.** The diagnostic performance based on different input parameters and models.

| Forecasting Methods | Photosynthetic Parameters | | Chlorophyll Fluorescence Parameters | | SIF Parameters | | PCs | | DCPs | |
|---|---|---|---|---|---|---|---|---|---|---|
| | Acc | RMSE | Acc | RMSE | Acc | RMSE | Acc | RMSE | Acc | RMSE |
| MLP | 83.33% | 0.423 | 83.33% | 0.599 | 91.67% | 0.323 | 100% | 0.224 | 100% | 0.172 |
| RF | 87.5% | 0.334 | 91.67% | 0.344 | 100% | 0.083 | 95.83% | 0.214 | 100% | 0.136 |
| SVM | 83.33% | 0.5 | 95.83% | 0.25 | 79.17% | 0.935 | 83.33% | 0.5 | 87.5% | 0.433 |
| MLR | 95.83% | 0.317 | 75% | 0.734 | 83.33% | 0.539 | 100% | 0.315 | 100% | 0.292 |
| Overall accuracy | 87.5% | | 86.46% | | 88.54% | | 95% | | 97% | |

Note: MLP: Multi-layer perceptron; RF: Random forest; SVM: Support vector machines; MLR: Multiple linear regression; Photosynthetic parameters: $G_s$, WUE, $C_i$, $L_s$, $Pn_{max}$, AQE; Chlorophyll fluorescence parameters: ABS/RC, $TR_0$/RC, $ET_0$/RC, $DI_0$/RC, $RE_0$/RC, $\psi_0$, $\varphi P_0$, $\varphi E_0$, $\varphi D_0$, $\varphi R_0$, $PI_{abs}$; SIF parameters: $aSIF_{683}$, $aSIF_{742}$, $rSIF_{687}$, $rSIF_{742}$, $rSIF_{760}$, $rSIF_{763}$; PCs: Principal components extracted based on principal component analysis (PCA), including WUE, $Pn_{max}$, $\varphi P_0$, $RE_0$/RC, $aSIF_{742}$ and $rSIF_{760}$; DCPs: Distance characteristic parameters extracted based on mean-centered standardized distance (MCSD), including $C_i$, AQE, $\varphi R_0$, $PI_{abs}$, $aSIF_{742}$, and $rSIF_{760}$.

**Table 7.** MLR models for diagnosis of low-temperature stress levels in strawberry seedlings.

| Variables | Multiple Regression Equation | $R^2$ |
|---|---|---|
| Photosynthetic parameters | $y = -4.356 * G_s - 0.156 * WUE + 0.008 * C_i + 0.327 * L_s + 0.035 * Pn_{max} - 13.967 * AQE - 0.39$ | 0.977 ** |
| Chlorophyll fluorescence parameters | $y = -0.061 * \frac{ABS}{RC} + 0.644 * \frac{TR_0}{RC} - 1.036 * \frac{ET_0}{RC} - 1.248 * \frac{RE_0}{RC} + 0.089 * \frac{DI_0}{RC} + 5.145 * \psi_0 + 9.063 * \varphi P_0 - 15.755 * \varphi E_0 + 0.652 * \varphi R_0 - 0.079 * PI_{abs} + 0.109$ | 0.932 ** |
| SIF parameters | $y = -0.838 * aSIF_{683} + 3.684 * aSIF_{742} - 26.465 * rSIF_{687} + 3.623 * rSIF_{742} - 50.295 * rSIF_{760} - 17.648 * rSIF_{763} + 3.646$ | 0.933 ** |
| PCs | $y = -0.257 * WUE - 0.015 * Pn_{max} - 0.271 * \varphi P_0 + 0.432 * \frac{RE_0}{RC} + 3.152 * aSIF_{742} - 40.1941 * rSIF_{760} + 3.102$ | 0.966 ** |
| DCPs | $y = 0.009 * C_i - 14.863 * AQE + 0.26 * \varphi R_0 - 0.167 * PI_{abs} + 2.669 * aSIF_{742} - 18.461 * rSIF_{760} - 0.633$ | 0.981 ** |

Note: ** indicates significance of difference at 0.01 level. MLP: Multi-layer perceptron; RF: Random forest; SVM: Support vector machines; MLR: Multiple linear regression; Photosynthetic parameters: $G_s$, WUE, $C_i$, $L_s$, $Pn_{max}$, AQE; Chlorophyll fluorescence parameters: ABS/RC, $TR_0$/RC, $ET_0$/RC, $DI_0$/RC, $RE_0$/RC, $\psi_0$, $\varphi P_0$, $\varphi E_0$, $\varphi D_0$, $\varphi R_0$, $PI_{abs}$; SIF parameters: $aSIF_{683}$, $aSIF_{742}$, $rSIF_{687}$, $rSIF_{742}$, $rSIF_{760}$, $rSIF_{763}$; PCs: Principal components extracted based on principal component analysis (PCA), including WUE, $Pn_{max}$, $\varphi P_0$, $RE_0$/RC, $aSIF_{742}$ and $rSIF_{760}$; DCPs: Distance characteristic parameters extracted based on mean-centered standardized distance (MCSD), including $C_i$, AQE, $\varphi R_0$, $PI_{abs}$, $aSIF_{742}$, and $rSIF_{760}$.

### 3.5.3. Construction of Diagnostic Decision Tree Model for Low-Temperature Stress in Strawberry Seedlings

The DCPs have not only shown a stronger differentiation between neighboring low-temperature stress levels but also provide comprehensive physiological information regarding the plants from different kinds of indicators, which highlights its advantages. Different indicators in DCPs have varying performances in distinguishing between different levels of low-temperature stress. To fully utilize the strengths of each distance characteristic parameter (DCP), the study employed them as the primary diagnostic criteria for their optimal distinguishable adjacent low-temperature stress levels, and serving as sub-conditions for diagnosing varying levels of low-temperature stress. By combining these sub-conditions and arranging them in a decision tree model, different stress levels can be evaluated and filtered layer by layer. The model is based on plant physiological indicators and has practical physical meanings for the non-destructive diagnosis of low-temperature stress. Table 8 shows the reliable confidence intervals of the six DCPs at different levels of low-temperature stress,

determined based on "Mean $\pm 2 \times$ SD". The threshold values of non-normally distributed indicators were tested and adjusted using the Bayesian discriminant rule. Discrimination rules for each level of low-temperature stress were determined after a detailed comparison.

**Table 8.** Confidence intervals of DCPs at different low-temperature stress levels.

| Stress Level | $C_i$ | AQE | $\varphi R_0$ | $PI_{abs}$ | $aSIF_{742}$ | $rSIF_{760}$ |
|---|---|---|---|---|---|---|
| Level 0 | $230.589 \pm 20.58$ | $0.045 \pm 0.009$ | $0.282 \pm 0.03$ | $10.727 \pm 2.148$ | $1.38 \pm 0.224$ | $0.15 \pm 0.019$ |
| Level 1 | $261.946 \pm 12.72$ | $0.008 \pm 0.002$ | $0.17 \pm 0.014$ | $5.388 \pm 0.8$ | $0.695 \pm 0.08$ | $0.083 \pm 0.009$ |
| Level 2 | $322.074 \pm 3.54$ | $0.002 \pm 0.0001$ | $0.148 \pm 0.011$ | $3.895 \pm 0.398$ | $0.55 \pm 0.004$ | $0.068 \pm 0.0008$ |
| Level 3 | $392.291 \pm 22.416$ | $0.0003 \pm 0.0005$ | $0.077 \pm 0.08$ | $0.458 \pm 0.736$ | $0.201 \pm 0.167$ | $0.026 \pm 0.018$ |

Note: Each value represents 'mean $\pm$ standard deviation (SD)'. Ci: Intercellular carbon dioxide concentration; AQE: Apparent quantum efficiency; $\varphi R_0$: The quantum yield of the terminal electron acceptors of PSI receptor; $PI_{abs}$: Absorption-based performance index; $aSIF_{742}$: Absolute SIF at 742 nm; $rSIF_{760}$: Relative SIF at 760 nm.

AQE as the first discriminant index aims to prioritize the removal of Level 0. Although AQE performs well in distinguishing Level 0&2 and Level 0&3, as shown in Table 5, its ability to distinguish Level 0&1 is limited. Therefore, the 'Yes' results in the first judgment layer may include all Level 0 and some Level 1 samples, and even a few Level 2 data may be mixed in. $\varphi R_0$ exhibits the best discriminative ability between Level 0&1, allowing for the separation of Level 0. While the probability of Level 2 entering this branch is small, we still included $aSIF_{742}$, which has good discriminative ability between Level 1&2, in the third layer of this branch to reduce the possibility of misclassifying Level 2 as Level 1. The samples that enter the 'No' branch in the first layer judgment include all of Level 3, most of Level 2, and a small number of Level 1 samples. To reduce the probability of misjudgment, a double-layer decision was made on the right branch using $rSIF_{760}$ and $PI_{abs}$ to screen out Level 3 samples. For the remaining Level 1 and Level 2 data, Ci with a high MCSD of 4.33 was used as the discriminant function to ensure the accuracy of these two stress-level judgments. By combining the discriminant criteria mentioned above, a decision tree model for diagnosing low-temperature stress in strawberry seedlings can be constructed as shown in Figure 8. The model achieved 100% accuracy in diagnosing the stress levels of all 24 validation samples.

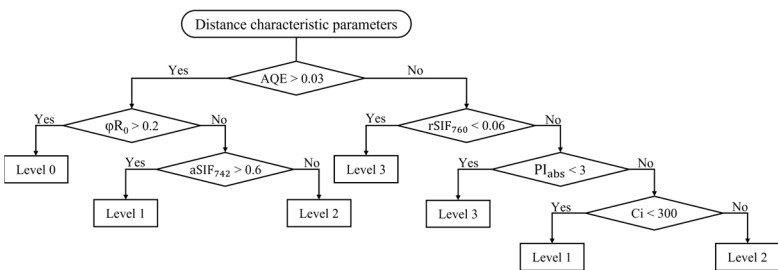

**Figure 8.** Decision tree for diagnosis of low-temperature stress level in strawberry seedlings.

## 4. Discussion

The physiological effects of low-temperature stress on plants depend on their own cold tolerance, growth stage, and exposure, as well as the intensity and duration of the low-temperature stress. [41]. Seedlings have weaker resistance to low temperatures during different growth stages. Low-temperature stress during the seedling stage can have adverse and far-reaching effects on crop growth, development, and yield quality [42]. This study investigated the low-temperature tolerance differences among strawberry varieties of different photoperiod types, under identical conditions controlling for plant growth stage, exposure, and duration, using continuous low-temperature gradients. The study proposed sensitive diagnostic indicators and methods for low-temperature stress diagnosis during the strawberry seedling stage. The results of this study may provide theoretical support for the strawberry industry in different climatic regions. In areas with lower temperatures,

short-day-type, seasonal strawberries may be more suitable for cultivation, while long-day-type, everbearing cultivars require reliable insulation measures. If strawberries have already been subjected to low-temperature stress, the diagnostic methods proposed in this study can assess the degree of strawberry damage without causing further harm and allow for timely remedial actions to minimize losses.

Low temperature primarily affects plant damage at the cellular membrane level, leading to a loss or reduction in selective permeability, and resulting in electrolyte leakage such as K+ and amino acids. The plasma membrane is considered the main site of injury in plant cells exposed to extracellular freezing [43]. Measuring the relative electrical conductivity (REC) of strawberry leaves reflects the degree of electrolyte leakage and reveals the level of plant damage under low-temperature stress. This study found that the relative electrical conductivity (REC) of strawberry leaves ranged from 20% to 85%. Between 0 and $-10\,^{\circ}$C, the cell membrane lipids of four strawberry varieties changed from the liquid crystalline phase to the gel phase, resulting in the disorder-to-order rearrangement of fatty acid chains and consequent cracking of the cell membrane [44]. The electrolyte leakage no longer significantly increased when the temperature dropped to $-15\,^{\circ}$C or lower. In this case, severe structural damage appeared in the leaves, with the degree of plasma membrane damage reaching a maximum and photosynthetic activity almost completely lost. This is consistent with the conclusion reached by Maughan et al. [45] after summarizing a large number of low-temperature stress experiments on strawberries, namely that unprotected strawberry plants suffer severe damage at $-9\,^{\circ}$C and die at $-12\,^{\circ}$C. We also noted that the low-temperature semi-lethal threshold ($LT_{50}$) for 'Selva' strawberries was measured at $-16.4\,^{\circ}$C in one study [46], which is likely due to insufficient treatment time. The shorter the duration of low-temperature stress, the less impact it has on plants, which may lead to a lower calculated $LT_{50}$ value.

In terms of photosynthetic parameters, Gs and $Pn_{max}$ are more sensitive indicators of sub-low temperature stress than WUE, Ls, and AQE. The study found that WUE first increased and then decreased as temperature decreased, with the highest WUE for short-day-type and long-day-type strawberries occurring at $5\,^{\circ}$C and $10\,^{\circ}$C, respectively. This may be due to the fact that water consumption per unit of time by plants decreases in sub-low temperature environments, leading to an increase in WUE. Combining Ci and Ls analysis can effectively determine whether the reduction in plant photosynthetic capacity is due to stomatal limitation or non-stomatal limitation factors [47]. We found the decrease in $Pn_{max}$ during the entire cooling process occurred due to both stomatal and non-stomatal factors. Ci continued to decrease during the cooling process that did not fall below $0\,^{\circ}$C, while Ls showed the opposite trend, indicating that the decrease in photosynthetic rate caused by temperatures above $0\,^{\circ}$C is due to stomatal factors. When the temperature further decreased, Ci increased significantly at $-5\,^{\circ}$C, while Ls decreased significantly, indicating that the influence of temperatures below $0\,^{\circ}$C on photosynthesis is due to non-stomatal factors, such as RuBP carboxylation limitation and photochemical activity limitation, which hinder $CO_2$ utilization and lead to an accumulation of intercellular $CO_2$ [48].

Transient chlorophyll fluorescence induction kinetics curve (OJIP transient) characteristics of four tested strawberry varieties revealed that the energy transfer of the PSII reaction center and OEC activity in different photoperiod types of strawberry will not be negatively affected in a sub-low temperature environment not lower than $5\,^{\circ}$C. Short-day-type strawberries exhibit stress characteristics at lower temperatures compared to long-day-type varieties. In terms of monitoring the stress of performance parameters of the PSII reaction center, it was found that in sub-low temperature environments, ABS/RC and $TR_0$/RC first increased and then decreased with decreasing temperature, while $ET_0$/RC showed an overall decreasing trend. The increase in $TR_0$/RC was lower than that in ABS/RC, and the temperature at which it reached its peak was also higher. It indicated that the ability to capture energy was less than the energy absorbed by the reaction center, but the absorbed light energy could not be used to promote electron transfer, leading to insufficient energy to maintain photosynthesis, forming an 'energy trap'. $PI_{abs}$ is an important indicator for

evaluating the efficiency of photosynthetic reactions and reflecting the impact of stress on photosynthetic structures [49]. In a study of 30 soybean varieties, Strauss et al. [50] pointed out that $PI_{abs}$ can not only reflect the differences in cold resistance between different varieties, but also characterize the differences in the ability of reaction centers to absorb light energy. The performance of $PI_{abs}$ in this study confirms this conclusion. $PI_{abs}$ not only performs well in distinguishing stress levels, but also reflects the differences between short-day-type and long-day-type strawberries under low-temperature stress. Compared with long-day-type strawberries, the $PI_{abs}$ of short-day-type cultivars decreases more slowly under low-temperature stress, indicating a stronger adaptation to low temperatures.

This study innovatively introduced solar-induced chlorophyll fluorescence (SIF) into the leaf-scale diagnosis of cold stress. SIF has the advantage of not requiring artificial light sources and is suitable for the non-destructive detection of plant photosynthetic physiological changes in natural environments. Recently, researchers have used SIF satellite inversion data to estimate vegetation productivity [51], potential evapotranspiration [52], and drought severity [53], and some studies have quantitatively calculated SIF for diseases [18] and chlorophyll content [16] at the leaf scale, but few researchers to date have explored SIF changes in plants under different degrees of low-temperature stress. In this study, SIF was used as the third type of data for low-temperature stress diagnosis, in conjunction with photosynthetic parameters and transient chlorophyll fluorescence parameters. SIF's potential for stress diagnosis at the leaf scale was investigated based on its absolute and relative spectral features. The results showed that the combination of $aSIF_{742}$ and $rSIF_{760}$ was the best match for differentiating low-temperature stress levels using SIF data. It is worth noting that different batches of samples observed under varying solar irradiance could affect absolute SIF, but this did not appear to affect its status as an important factor for diagnosing strawberry cold damage.

Distinguishing adjacent levels of low-temperature stress is crucial but challenging, as physiological indicators of plants tend to be similar and often overlap. Compared to principal component analysis (PCA), mean-centered standardized distance (MCSD) focuses on differences between samples, resulting in less redundant information and better performance in distinguishing low-temperature stress levels. While feature indicators selected by PCA are less diagnostic than distance characteristic parameters (DCPs) for low-temperature stress levels, they are still significantly better than using one single type of data. Although the parameters of photosynthesis, transient chlorophyll fluorescence, and SIF all describe plant light performance, the information they provide is not entirely redundant and still has complementary space. In addition, the combination of different types of parameters can effectively avoid the randomness that may exist when using a single type of data. This research combines the decision trees model with DCPs. Although decision tree is a classic classification method in geosciences, their characteristic of sequential judgment based on key indicators and step-by-step screening is similar to the relationship between plant physiological indicators and stress levels. This has led to the development of a non-destructive diagnostic procedure for low-temperature stress with plant physiological significance in a concise and clear manner.

The conclusions of this study are based on short-day-type strawberry cultivars 'Toyonoka' and 'Red Face', and long-day-type varieties 'Selva' and 'Sweet Charlie'. Whether the low-temperature resistance of other strawberry cultivars matches the patterns summarized in this study or if the threshold values of the low-temperature stress diagnosis model for strawberry seedlings need to be further adjusted and optimized, will require additional stress experiments with more strawberry cultivars and samples in the future.

## 5. Conclusions

Under 12 h, low-temperature stress of varying degrees, the short-day-type strawberries represented by 'Toyonoka' and 'Red Face' exhibited better tolerance to low temperatures during the seedling stage than the long-day-type varieties represented by 'Selva' and 'Sweet Charlie'. This was reflected in terms of cell damage and photosynthetic activity. The semi-

lethal temperatures for 'Toyonoka', 'Red Face', 'Selva', and 'Sweet Charlie' were $-5.48$ °C, $-7.67$ °C, $-2.13$ °C, and $-3.38$ °C, respectively. The cold resistance of the four strawberry cultivars ranked from strong to weak is 'Red Face', 'Toyonoka', 'Sweet Charlie', and 'Selva'. Strawberries of different photoperiodic types start experiencing low-temperature stress at temperatures below 5 °C and all suffer severe damage when exposed to temperatures within the range of 0 to $-10$ °C. Short-day cultivars have a lower critical temperature for low-temperature damage than long-day cultivars. The photosynthetic performance of long-day-type strawberries begins to decline significantly at 0 °C, while the photosynthetic ability of the short-day-type cultivars only significantly decreases at $-5$ °C. When the temperature drops to $-10$ °C, the physiological activity of different photoperiod types of strawberries is almost completely lost.

Mean-centered standardized distance is a good method for extracting indicators of stress level distinguishability. Ci, AQE, $\varphi R_0$, $PI_{abs}$, $aSIF_{742}$, and $rSIF_{760}$ showed great ability to distinguish different levels of low-temperature stress, and combining different physiological parameters is more comprehensive in describing the stress state of vegetation than relying solely on single indices of a certain type. Based on the above, the study constructed a non-destructive diagnosis decision tree model and established five non-destructive diagnosis formulas for low-temperature stress on strawberry seedlings. This study provides scientific references and a decision-making approach for the selection of strawberry cultivars, as well as the prevention and non-destructive diagnosis of low-temperature stress during the seedling stage in practical production.

In future work, we plan to expand the number and variety of strawberry cultivars studied, investigate the physiological significance between different types of non-destructive stress diagnostic indicators, and further enrich the knowledge regarding low-temperature stress on strawberry seedlings based on this study.

**Author Contributions:** Conceptualization, N.J. and Z.Y.; methodology, N.J., Z.Y. and C.L.; software, N.J.; validation, N.J. and Z.Y.; formal analysis, N.J.; investigation, N.J., H.Z. and J.X.; resources, N.J.; data curation, N.J., H.Z. and J.X.; writing—original draft preparation, N.J. and Z.Y.; writing—review and editing, H.Z., J.X. and C.L.; supervision, Z.Y. and C.L; project administration, N.J. and Z.Y.; funding acquisition, Z.Y. All authors have read and agreed to the published version of the manuscript.

**Funding:** This work was funded by the National Natural Science Foundation of China (NO.42275200).

**Data Availability Statement:** The original contributions presented in the study are included in the article. Further inquiries can be directed to the corresponding author.

**Conflicts of Interest:** The authors declare no conflict of interest.

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
