# Peer review of "Effect of Low Temperature on Photosynthetic Physiological Activity of Different Photoperiod Types of Strawberry Seedlings and Stress Diagnosis"

_agronomy, doi:10.3390/agronomy13051321_

Round 1
Reviewer 1 Report
Manuscript Title: Effect of Low Temperature on Photosynthetic Physiological Activity of Different Photoperiod Types of Strawberry Seedlings and Non-destructive Diagnosis.
The title and subject of the manuscript are very interesting from the methodological and practical point of view, suitable and adequate. The scientific content contributes to the space in which it develops.
The analysis of the published data was provided with a sufficient level of scientific novelty. The abstract of the paper is factual concrete, realistic, and understandable.
The introduction provides a good understanding of the subject and its importance, with a significant quantity of information. Theoretical and practical reasons for the experiments are very reasonable.
The materials and methods are written clearly and in detail for the reader to understand.
The results were described nicely and accurately and discussed very well according to my knowledge.
There are some minor corrections that I have noticed that may improve the standard of the manuscript.
I recommend that this manuscript be published after following the corrections suggested in the attached file.
I hope my comments improve the quality of your manuscript
Best regards

Author Response
Response to Reviewer 1 Comments
Manuscript ID: agronomy-2377956
Esteemed reviewer:
Greetings!
We sincerely appreciate your careful review of our manuscript. Your professional and meticulous suggestions have been carefully considered and addressed. Your valuable comments have helped us to improve the manuscript.
In this cover letter, we provide explanations to the main issues raised by you. Other issues such as formatting, typos, and grammar errors have been detailed in the revised manuscript used the
“Track Changes” in the attached file (which includes revisions made according to all the reviewers' comments). We have highlighted the main issues you raised with comments.
We express our gratitude for your hard work on our manuscript. It is our pleasure to work with you. If there are any further areas for improvement, we will continue to work , to make our contribution to agriculture.
Wish you a successful career and a happy life!
Best regards~
Sincerely yours, Author of agronomy-2377956
Point 1: Use other words not in the title, that is let your paper appear easy with readersfor example: OJIP; PSII reaction center; JIP-test. Use alphabetical order
Response 1: According to your suggestion, we have re-adjusted the keywords to avoid repetition with the title and re-ordered them alphabetically. The new keywords are: Fragaria × ananassa Duch; Long-day varieties; Low temperature stress; Non-destructive diagnosis; OJIP transient; Photosynthesis; Relative electrical conductivity; Seedling stage; Short-day varieties; SIF.
Point 2: Will be better to insert a reference about strawberry not cucumber. / Add this reference related to OJIP test for abiotic stress. / These references so old replace them with xxx.
Response 2: We have added and modified the more appropriate literature recommended by you according to your suggestion. However, we could not locate the DOI address (https://doi.org/10.32615/ps.2019.150) you provided on both Google Scholar and the journal's official website. The highest serial number in 2019 seems to be around 130, and there is no record of 150. Therefore, we have kept reference [29] for now. Although it is relatively old, it is the most classic and highly cited JIP-test guide. We have successfully replaced the other reference we originally cited with one of the references you recommended. But we have smoothly replaced and added the other literature recommended by you.
Point 3: In the aim of the study, the hypothesis assumed by the authors should be added.
Response 3: According to your suggesstion, we have added the hypothesis proposed by the author. “Different photoperiodic strawberry varieties are suitable for growth in different seasons, which leads us to speculate that there may be differences in cold adaptation be-tween short-day and long-day types.”
Point 4: Some researchers need to follow you to product strawberry seedlings please add more details.
Response 4: We have supplemented the specific characteristics of the seedlings used in the experiment as per your suggestion. “During the cultivation period, all test seedlings were subjected to uniform water and nu-trient management until the start of the low temperature treatment. The soil was fertilized with nitrogen (urea, 46% N, 150 kg/ha), phosphorus (calcium superphosphate, 12% P2O5, 200 kg/ha), and potassium (potassium sulfate, 52% K2O, 250 kg/ha) as the basal fertilizer, without further fertilization since then. Water was supplied according to the ‘5-point sampling method’, with additional watering to saturation when the average soil moisture content reached approximately 60%. The watering was carried out between 16:00-18:00.”
Point 5: Why did the temperture and humidity change per a day?? You have used artificial climate chamber
Response 5: Because we are simulating the 24-hour environment changes in a greenhouse. The artificial climate chamber we used (PGC-FLEX, Conviron, Canada) has the function of adjusting temperature and humidity hourly. Therefore, we set up dynamic changes in temperature and humidity throughout the day to simulate the actual growth environment.
Point 6: Explain more when you use 3000 and 200 μmol s−1. Depend my knowldge should be:(3000 μmolm-2 s-1 and 200 ms duration)
Response 6: As per your suggestion, we have provided a specific description of the duration of saturation pulses and the duration of light for light-dark transitions. “The saturation pulse intensity was set at 3000 with a duration of 1s. The actinic light intensity was set at 200 with a duration of 9s. Each single-point measurement lasted for 210 seconds, with alternating saturation pulse and actinic light every 10 seconds in each cycle.”
Point 7: The result of aFSR is absolute SIF (aSIF)??????
Response 7: Here we failed to introduce clearly before, aFSR is the method for inverting SIF, and the inverted SIF is absolute SIF (aSIF), which looks quite similar to each other.
Point 8: When I started reading the results, I thought the authors had mixed results with discussion. There are many sentences to discuss the results in the results section. I think it is not needed as long as there is a separate discussion section.
Response 8: We have strictly followed your suggestions in the Results section, transferring some content from the Results section to the Discussion section, and extensively deleting statements that are redundant and do not contribute to the development of the article, which can be inferred from the figures and tables, to highlight the key findings.
Point 9: When I started reading the results, I thought the authors had mixed results with discussion. There are many sentences to discuss the results in the results section. I think it is not needed as long as there is a separate discussion section.
Response 9: In the Results section, we have strictly followed your suggestions and extensively deleted statements that could be inferred from figures and tables and do not contribute to the development of the article, in order to highlight the key findings.
Point 10: Please increase the resolution of figures. I suggest writing the significance in one box so that the reader can see the significant differences for each curve separately. As I have done in my research paper: https://journals.plos.org/plosone/article?id=10.1371/journal.pone.0257745. I have done it by Excel, I hope you can do it using your program. If you couldn't you can change the color of curves to be clear (blach+ blue+ red+green).
Response 10: To maintain the consistency of the color tone of all figures throughout the article, we have chosen to follow one of your suggestions to present the significance level in a separate table below the figure, which indeed improved the readability. In addition, we have increased the resolution of all figures in the article to 1200ppi, and please refer to the attached file for the improved quality.
Point 11: ‘ ,respecticely’ & ‘P<0.05’
Response 11: We have checked the entire manuscript and added a comma before 'respectively' and italicized 'P<0.05'.
Point 12: I'm not sure if this sentence should be kept or deleted. (Line 456)
Response 12: We were also very conflicted about this. Once without necessary explanations, it might be difficult for readers to understand the meanings of ΔWok and ΔWoj in the figure, as the values of several points on the ΔW curve do not have the clear referential meanings like other simple parameters.
Point 13: Add future plan in the conclusion section and try to decrease it, you don't have to add all results in the conclusion.
Response 13: Based on the review experts's suggestions, we significantly reduced the content in the conclusion section that is not directly related to the key findings, and added our future research plans at the end. “In future work, we plan to expand the number and variety of strawberry cultivars studied, investigate the physiological significance between different types of non-destructive stress diagnostic indicators, and further enrich the knowledge of low-temperature stress on strawberry seedlings based on this study.”

Reviewer 2 Report
General recommendations and questions
Title
The title of the article is not correct.
“….non-destructive diagnosis”. Stress diagnosis? How low temperatures can affect non-destructive diagnosis? Correct the title.
Keywords: Repeating the same terms in the title and keywords is not recommended.
You can use, for example - low temperature stress, the Latin name of strawberries, short-day varieties, long-day varieties etc.
Introduction
The main aim of the research should be clearly formulated. The goal of the study cannot be to perform or achieve several objectives.
Materials and Methods
If the full Latin name for the species has already been given once, there is no need to use it further in the text.
Line 86. Strawberry varieties of different photoperiod types?
Line 91. “1.5-gallon pots” It is not in the SI system. The description of the pots should be clarified.
The methodology of the experiment is not entirely clear. How many pots for each variant, how many times has the low temperature treatment been performed? Please describe the experimental setup in more detail.
Line 111-112. The text is incomprehensible; sentences are missing verbs.
Line 119-120. Unclear. Leaf fragments were transferred? Water was added? Why - subjected to vacuum suction for 0 min?
Check that all indicators are explained in the equations. This is not done for several equations, for example 2 and 4.
Since the article uses a lot of abbreviations, check that they are all explained. To make it easier to follow the content of the article, perhaps you should give full names at the beginning of the chapters also later.
Results
The results section is too long. This is understandable, because a lot of indicators have been analysed. However, it is preferable to avoid retelling the data that we can see in the Figures and Tables. Be as focused as possible.
It is necessary to avoid discussing the data in the Results section (for example Line 219-221). This is for discussion part.
You also use synonyms extensively as everbearing cultivars, long-day type cultivars even in the one sentence (Line 256-259). It is sometimes confusing.
Line 272. “At 0°C, the average Gs of seasonal cultivars was 0.09, which was lower than that of everbearing cultivars 0.07.” Why lower?
Line 306-308. “At the turning point of the Ci and Ls trends, which is 0°C, the average Ci and Ls values of seasonal and everbearing strawberries with a difference of 23.32 μmol mol−1 and 0.09 μmol mol−1 respectively.” Unclear.
Short description of OJIP phases is recommendable. For example: O minimal fluorescence, P max fluorescence etc.
PCA analysis was used in data analysis but is not mentioned in the Materials and Methods.
What is RMSE in Table 6? Root-mean-square deviation? There is no explanation.
Line 618-620. Unclear, please rephrase.
Table 8. There is no explanation for the abbreviations used in the table.
Figures.
All Figures should indicate which are the short-day varieties and which are the long-day varieties. This is not done in all Figures.
The Figures are unusually oriented, placing the control variant on the right. Usually we look from left to right, but here we have to look from right to left. I don't think it's convenient. If this Figure design is related to the temperature scale, then why did you use the notation CK instead of 20 degrees Celsius, thus consistently avoiding the double distance of 10 degrees. The other treatments could also have been simple designations. However, these are just my thoughts on the Figures, everything is understandable.
Discussion
Do not use the term habit instead of the term habitat.
Line 657. It is highly questionable that this study focuses on plant habitats and growth stages. You haven't studied anything like that!
It is recommended to emphasize more the potential possibility of applying the research results practically. This aspect has been minimally developed, but the results of this study could be successfully used.
Conclusions
The Conclusion part should be shorter and more focused.
Line 756. Unclear sentence.
Line 773. Do not start any sentence with "And".
Concluding remarks. In general, the article is interesting, contains new knowledge and has rich data material. I recommend accepting this article in Agronomy after minor revision.
The article is written in generally good English, is clearly understandable. However, there are some gaps: sentences are sometimes missing a verb, word order should be checked, and sentences that are too long should be avoided where possible.
Author Response
Response to Reviewer 2 Comments
Manuscript ID: agronomy-2377956
Esteemed reviewer:
Greetings!
We sincerely appreciate your careful review of our manuscript. Your professional and meticulous suggestions have been carefully considered and addressed. Your valuable comments have helped us to improve the manuscript.
In this cover letter, we provide explanations to the main issues raised by you. Other issues such as formatting, typos, and grammar errors have been detailed in the revised manuscript used the
“Track Changes” in the attached file (which includes revisions made according to all the reviewers' comments). We have highlighted the main issues you raised with comments.
We express our gratitude for your hard work on our manuscript. It is our pleasure to work with you. If there are any further areas for improvement, we will continue to work , to make our contribution to agriculture.
Wish you a successful career and a happy life!
Best regards~
Sincerely yours, Author of agronomy-2377956
【Tittle】&【Keywords】
Point 1: “….non-destructive diagnosis”. Stress diagnosis? How low temperatures can affect non-destructive diagnosis? Correct the title. & Repeating the same terms in the title and keywords is not recommended.
Response 1: According to your suggestion, we have made modifications to the article title and keywords. The current version of the title is "Effect of Low Temperature on Photosynthetic Physiological Activity of Different Photoperiod Types of Strawberry Seedlings and Stress Diagnosis", and the keywords are "Fragaria × ananassa Duch; Long-day varieties; Low temperature stress; Non-destructive diagnosis; OJIP transient; Photosynthesis; Relative electrical conductivity; Seedling stage; Short-day varieties; SIF".
【Introduction】
Point 2: The main aim of the research should be clearly formulated. The goal of the study cannot be to perform or achieve several objectives.【Line 89】
Response 2: According your suggestion, we have re-write a concise summary of the main goals of this article. “Thus, the study aims to investigate the variations and discrepancies in the photosynthetic physiological activity of short-day type and long-day type strawberry seedlings under low temperature stress, and to establish a non-destructive diagnostic method for low-temperature stress degree during the seedling stage by key indicators derived from photosynthetic parameters and chloro-phyll fluorescence characteristics .”
【Materials and Methods】
Point 3: “1.5-gallon pots” It is not in the SI system. The description of the pots should be clarified.The methodology of the experiment is not entirely clear. How many pots for each variant, how many times has the low temperature treatment been performed? Please describe the experimental setup in more detail.【Line 107】
Response 3: According your suggestion, here, we have provided detailed information on the cultivation process of the seedlings used in the experiment and supplemented information about the number of plants and pots used in the study. “During the cultivation period, all test seedlings were subjected to uniform water and nu-trient management until the start of the low temperature treatment. The soil was fertilized with nitrogen (urea, 46% N, 150 kg/ha), phosphorus (calcium superphosphate, 12% P2O5, 200 kg/ha), and potassium (potassium sulfate, 52% K2O, 250 kg/ha) as the basal fertilizer, without further fertilization since then. Water was supplied according to the ‘5-point sampling method’, with additional watering to saturation when the average soil moisture content reached approximately 60%. The watering was carried out between 16:00-18:00. Each treatment involved three strawberry plants, resulting in a total of 24 plants for each variety. A total of 96 strawberry seedlings were utilized for the four varieties with 96 sep-arate pots .”
Point 4: Unclear. Leaf fragments were transferred? Water was added? Why - subjected to vacuum suction for 0 min?【Line 153】
Response 4: Previously, it was our negligence that led to an incomplete description of this section. Now we have revised it according to your reminder. ”Weigh 1.5 g of chopped leaves and place them into a 25 mL conical flask. Add 20 mL of deionized water to the flask and vacuum the mixture for 20 minutes until the leaves settle at the bottom .”
Point 5: Check that all indicators are explained in the equations. This is not done for several equations, for example 2 and 4.& Since the article uses a lot of abbreviations, check that they are all explained. To make it easier to follow the content of the article, perhaps you should give full names at the beginning of the chapters also later.
Response 5: According to your reminder, we have added text annotations below all the figures and tables in the article to introduce their respective indicators. This includes adding an introduction to the long-day and short-day strawberry varieties below each figure and table. In addition, we have added the full names before the first appearance of each abbreviation in every section.
【Results】
Point 6: The results section is too long. This is understandable, because a lot of indicators have been analysed. However, it is preferable to avoid retelling the data that we can see in the Figures and Tables. Be as focused as possible. & It is necessary to avoid discussing the data in the Results section (for example Line 219-221). This is for discussion part. 【Line 271】
Response 6: We have strictly followed your suggestions in the Results section, transferring some content from the Results section to the Discussion section, and extensively deleting statements that are redundant and do not contribute to the development of the article, which can be inferred from the figures and tables, to highlight the key findings.
Point 7: “At the turning point of the Ci and Ls trends, which is 0°C, the average Ci and Ls values of seasonal and everbearing strawberries with a difference of 23.32 μmol mol−1 and 0.09 μmol mol−1 respectively.” Unclear.【Line 374】
Response 7: This part was not described clearly before, and now we have rephrased it. “At 0°C, both short-day type and long-day type strawberries showed a turning point in the trends of Ci and Ls. The average Ci of short-day type strawberries was 23.32 μmol mol^(-1) higher than that of long-day type varieties, while the average Ls of long-day type strawberries was 0.09 μmol mol^(-1) higher than that of short-day type varieties .”
Point 8: Short description of OJIP phases is recommendable. For example: O minimal fluorescence, P max fluorescence etc.【Line 406】
Response 8: According to your suggestion, we have provided a brief introduction to the different phases of the OJIP trainsient. “The OJIP transient can be divided into four phases: the O phase, representing the initial fluorescence; the J phase, representing the increase in fluorescence signal due to the ac-cumulation of oxidized QA and electron transfer beyond QA; the I phase, representing the fluorescence inflection point where QA is converted to QB and electrons transfer to PSI; and the P phase, representing the point of maximum fluorescence signal.” (It may look a bit lengthy, but the use of the terms Fi and Fj can be somewhat complicated. T_T)
Point 9: PCA analysis was used in data analysis but is not mentioned in the Materials and Methods. & What is RMSE in Table 6? Root-mean-square deviation? There is no explanation.
Response 9: According to your suggestion, we have supplemented the introduction of the PCA method in the 'Materials and Methods' section, and also added the meanings of the model accuracy evaluation indicators used in the experiment.【Line 233-247】
【Figures】
Point 10: The Figures are unusually oriented, placing the control variant on the right. Usually we look from left to right, but here we have to look from right to left. I don't think it's convenient. If this Figure design is related to the temperature scale, then why did you use the notation CK instead of 20 degrees Celsius, thus consistently avoiding the double distance of 10 degrees. The other treatments could also have been simple designations. However, these are just my thoughts on the Figures, everything is understandable.
Response 10: According to the suggestions of the two experts, we replaced "CK" with the specific temperature of "20°C" and made consistent revisions to the text and numbers throughout the figure. After removing the "CK" symbol, it seems more appropriate to have the axis go from negative to positive values from left to right. Furthermore, since the temperature interval between 10°C and 20°C is 10°C rather than 5°C, we added a breakpoint between these two temperatures to improve the visual appearance of the figure. Additionally, we have added a table below the figure to show the letters representing significant differences separately.
Point 11: All Figures should indicate which are the short-day varieties and which are the long-day varieties. This is not done in all Figures.
Response 11: According to your suggestion, we have added detailed annotations in each figure and table to indicate which are the long-day types and which are the short-day types. In addition, you mentioned that synonyms can be confusing, so in this article, we have replaced all occurrences of "seasonal" and "everbearing" with "short-day type" and "long-day type," respectively, to better match the title of the article.
【Discussion】
Point 12: It is highly questionable that this study focuses on plant habitats and growth stages. You haven't studied anything like that! It is recommended to emphasize more the potential possibility of applying the research results practically. This aspect has been minimally developed, but the results of this study could be successfully used.【Line 767】
Response 12: Due to our poor phrasing, we did not mean to express the idea of 'habitat', but rather wanted to illustrate several aspects that cause damage to plants, i.e., 'vulnerable organs, inducing factors, and exposure'. We hoped to give a detailed introduction to the various aspects that lead to plant damage, and then selected 'low temperature intensity' and 'cold resistance of plants' for experimentation. According to your suggestion, we have re-explained this part and added potential application scenarios for this research. ” This study investigated the low-temperature tolerance differences among strawberry varieties of different photoperiod types, under identical conditions controlling for plant growth stage, exposure and duration, using continuous low-temperature gradients. The study proposed sensitive diagnostic indicators and methods for low-temperature stress diagnosis during the strawberry seedling stage. The results of this study may provide theoretical support for the strawberry industry in different climatic regions. In areas with lower temperatures, short-day type, seasonal strawberries may be more suitable for cultivation, while long-day type, everbearing cultivars require reliable insulation measures. If strawberries have already been subjected to low-temperature stress, the diagnostic methods proposed in this study can assess the degree of strawberry damage without causing further harm and allow for timely remedial actions to minimize losses.”
【Conclusions】
Point 13: The Conclusion part should be shorter and more focused.【Line 886】
Response 13: Based on the your suggestions, we significantly reduced the content in the conclusion section that is not directly related to the key findings, and added our future research plans at the end.
